# Fast Value Tracking for Deep Reinforcement Learning

**Frank Shih**
Department of Statistics
Purdue University
West Lafayette, IN 47907, USA
`shih37@purdue.edu`

**Faming Liang**
Department of Statistics
Purdue University
West Lafayette, IN 47907, USA
`fmliang@purdue.edu`

## Abstract

Reinforcement learning (RL) tackles sequential decision-making problems by creating agents that interacts with their environment. However, existing algorithms often view these problem as static, focusing on point estimates for model parameters to maximize expected rewards, neglecting the stochastic dynamics of agent-environment interactions and the critical role of uncertainty quantification. Our research leverages the Kalman filtering paradigm to introduce a novel and scalable sampling algorithm called Langevinized Kalman Temporal-Difference (LKTD) for deep reinforcement learning. This algorithm, grounded in Stochastic Gradient Markov Chain Monte Carlo (SGMCMC), efficiently draws samples from the posterior distribution of deep neural network parameters. Under mild conditions, we prove that the posterior samples generated by the LKTD algorithm converge to a stationary distribution. This convergence not only enables us to quantify uncertainties associated with the value function and model parameters but also allows us to monitor these uncertainties during policy updates throughout the training phase. The LKTD algorithm paves the way for more robust and adaptable reinforcement learning approaches.

## 1 Introduction

Over the last decade, RL has achieved remarkable successes across a diverse array of tasks, including robotics (Kormushev et al., 2013), video games (Silver et al., 2016), bidding strategies (Jin et al., 2018), and ridesharing optimization (Xu et al., 2018b). As a mathematical model, RL solves sequential decision-making problems by designing an agent that interacts with the environment, the goal is to learn an optimal policy that maximizes the expected total reward for the agent. Prominent value-based algorithms, including Temporal-difference (TD) learning (Sutton, 1988), State–action–reward–state–action (SARSA) (Sutton & Barto, 2018), and Q-learning, aim to derive an optimal policy through learning values of states (or Q-values). Traditionally, these methods treat the state value (or Q-value) as a deterministic function, focusing on calculating point estimates of model parameters, thereby overlooking the inherent stochasticity in agent-environment interactions.

In the context of RL, a fair algorithm should exhibit the features: (i) *Uncertainty quantification*, which addresses the stochastic nature of the agent-environment interactions, thereby enhancing the robustness of the learned policy; (ii) *Dynamicity*, which considers the dynamics of the agent-environment interaction system, thereby enhancing the practicality of the RL technique; (iii) *Nonlinear approximation*, which employs, for example, a deep neural network to approximate the value function, thereby broadening the algorithm's applicability; (iv) *Computational efficiency*, which is scalable with respect to the model dimension and training sample size, facilitating online learning. Therefore, in RL, it is more suitable to treat values or model parameters as random variables rather than fixed unknowns, focusing on tracking dynamic changes rather than achieving point convergence during the policy learning process.

To achieve these goals, the Kalman Temporal Difference (KTD) framework has been studied for RL in the literature, as seen in references such as e.g., Geist & Pietquin (2010), Tripp &

Shachter (2013), and Shashua & Mannor (2020). In these studies, values or their parameters are treated as random variables, and the focus is on the tracking property of the policy learning process. Specifically, KTD conceptualizes RL as a state-space model:

$$
\begin{aligned}
\theta_t &= \theta_{t-1} + w_t, \\
\boldsymbol{r}_t &= h(\boldsymbol{x}_t, \theta_t) + \eta_t,
\end{aligned}
\tag{1}
$$

where $\theta_t \in \mathbb{R}^p$ denotes the parameters at time step $t$ with dimension $p$, $w_t \in \mathbb{R}^p$ and $\eta_t \in \mathbb{R}^n$ denote two independent multivariate Gaussian vectors, $\boldsymbol{x}_t$ denotes a set of states and actions collected at time step $t$, $\boldsymbol{r}_t \in \mathbb{R}^n$ denotes a vector of rewards, $n$ denotes the number of samples, and $h(\cdot)$ is a function to be defined in Section 2.2. Within the framework of state-space models, the top equation in (1) is reffered to as the state evolution equation, while the bottom equation is known as the measurement equation. Under the normality assumption and for a linear measurement equation, where $h(\boldsymbol{x}, \theta)$ is a linear function of $\theta$, the Kalman filter (Kalman, 1960) is able to iteratively update the mean and variance estimates for $\theta_t$ conditioned on the rewards $(\boldsymbol{r}_t, \boldsymbol{r}_{t-1}, \ldots, \boldsymbol{r}_1)$, enabling a proper quantification of the uncertainty associated with the dynamic agent-environment interaction system. However, when $h(\boldsymbol{x}, \theta)$ becomes nonlinear, it necessitates the use of linearization techniques. Specifically, Geist & Pietquin (2010) employs Unscented Kalman Filter (UKF) (Wan & Van Der Merwe, 2000), while Shashua & Mannor (2020) utilize Extended Kalman Filter (EKF) Anderson et al. (1979) to approximate the covariance matrices of $\theta_t$. Unfortunately, both UKF and EKF becomes computationally inefficient for high-dimensional parameter spaces, a common scenario when employing large-scale neural networks to approximate $h(\cdot, \cdot)$. These filters require $O(p^2)$ additional space to store the covariance matrix and $O(np^2)$ for matrix multiplications at each iteration. Moreover, the linearization operation involved in these algorithms can degrade the accuracy of estimation. To address the limitations encountered by KTD, we reformulate RL as the following state space model:

$$
\begin{aligned}
\theta_t &= \theta_{t-1} + \frac{\epsilon_t}{2} \nabla_\theta \log \pi(\theta_{t-1}) + w_t, \\
\boldsymbol{r}_t &= h(\boldsymbol{x}_t, \theta_t) + \eta_t,
\end{aligned}
\tag{2}
$$

where $w_t \sim N(0, \epsilon_t I_p)$, $\pi(\theta)$ represents a prior density function we impose on $\theta$, and $\{\epsilon_t : t = 1, 2, \ldots\}$ is a positive sequence decaying to zero. Additionally, we propose to update $\theta_t$ using the Langevinized Ensemble Kalman Filter (LEnKF) algorithm (Zhang et al., 2023). With the formulation (2) and the LEnKF algorithm, we show in Section 3 that $\theta_t$ converges to a proper distribution as the learning horizon $t \to \infty$, enabling the uncertainty associated with the dynamic agent-environment interaction system to be properly quantified. Inclusion of the prior information in the state evolution equation generally robustifies the performance of the RL algorithm. For instance, when employing a large-scale deep neural network to approximate the function $h(\cdot, \cdot)$, selecting an appropriate $\pi(\cdot)$, such as a mixture Gaussian distribution, can lead to the sparsification of the neural network. This enhances the robustness of the learned policy according to the theory of sparse deep learning (Sun et al., 2022). Compared to existing KTD algorithms, the proposed algorithm can directly handle a nonlinear function $h(\cdot, \cdot)$ without the need for a linearization operator. The proposed algorithm enables fast value tracking at a complexity of $O(np)$ per iteration, scalable for large-scale neural networks. It also enhances memory-efficiency as it replaces the storage for the covariance matrix with particles, representing samples of $\theta$. It is worth noting that both the sample size $n$ and the number of particles retained during the algorithm's execution are typically significantly smaller than the parameter size $p$. Lastly, we extend the convergence theory of our proposed algorithm to include scenarios that utilize replay buffers, thereby expanding its applicability beyond the on-policy framework.

Compared to model (1), our new formulation imposes slightly more restrictions on the variability of $\theta$ through the prior $\pi(\theta)$, while still accounting for the dynamics of the system. However, these restrictions do not diminish the generality and adaptivity of the model (2), thanks to the universal approximation ability of deep neural networks that will be used in approximating $h(\cdot, \cdot)$ in this paper.

**Related Works** Bootstrapped DQN (Osband et al., 2016) and Quantile Regression DQN (Bellemare et al., 2017) also aim to learn the uncertainty estimates for the value function, but they are not formulated under the KTD framework.

## 2 BACKGROUND

### 2.1 MARKOV DECISION PROCESS

The standard RL procedure aims to learn an optimal policy from the interaction experiences between an agent and an environment, where the optimal policy maximizes the agent's expected total reward. The RL procedure can be described by a Markov Decision Process (MDP) represented by $\{\mathcal{S}, \mathcal{A}, \mathcal{P}, r, \gamma\}$, where $\mathcal{S}$ is set of states, $\mathcal{A}$ is a finite set of actions, $\mathcal{P} : \mathcal{S} \times \mathcal{A} \times \mathcal{S} \to \mathbb{R}$ is the state transition probability from state $s$ to state $s'$ by taking action $a$, denoted by $\mathcal{P}(s'|s, a)$, $r(s, a)$ is a random reward received from taking action $a$ at state $s$, and $\gamma \in (0, 1)$ is a discount factor. At each time step $t$, the agent observes state $s_t \in \mathcal{S}$ and takes action $a_t \in \mathcal{A}$ according to policy $\rho$ with probability $P_\rho(a|s)$, then the environment returns a reward $r_t = r(s_t, a_t)$ and a new state $s_{t+1} \in \mathcal{S}$. For a given policy $\rho$, the performance is measured by the state value function $V_\rho(s) = \mathbb{E}_\rho[\sum_{t=0}^\infty \gamma^t r_t | s_0 = s]$ and the state-action value function $Q_\rho(s, a) = \mathbb{E}_\rho[\sum_{t=0}^\infty \gamma^t r_t | s_0 = s, a_0 = a]$, which are called $V$-function and $Q$-function, respectively. Both functions satisfy the Bellman equation:

$$V_\rho(s) = \mathbb{E}_\rho[r(s, a) + \gamma V_\rho(s')],$$
$$Q_\rho(s, a) = \mathbb{E}_\rho[r(s, a) + \gamma Q_\rho(s', a')], \tag{3}$$

where $s' \sim \mathcal{P}(\cdot|s, a)$, $a \sim P_\rho(\cdot|s)$, $a' \sim P_\rho(\cdot|s')$, and the expectation $\mathbb{E}_\rho[\cdot]$ is taken over the transition probability distribution $\mathcal{P}$ for a given policy $\rho$.

### 2.2 KALMAN TEMPORAL DIFFERENCE (KTD) ALGORITHMS

Let $\boldsymbol{s}_t = (s_t^{(1)}, s_t^{(2)}, \ldots, s_t^{(n)})^T$, $\boldsymbol{a}_t = (a_t^{(1)}, a_t^{(2)}, \ldots, a_t^{(n)})^T$, and $\boldsymbol{r}_t = (r_t^{(1)}, r_t^{(2)}, \ldots, r_t^{(n)})^T$ denote, respectively, a vector of $n$ states, actions, rewards collected at time step $t$. Given the Bellman equation, the function $h(\boldsymbol{x}_t, \theta_t)$ in (1) can be expressed as

$$h(\boldsymbol{x}_t, \theta_t) = \begin{cases} V_{\theta_t}(\boldsymbol{s}_t) - \gamma V_{\theta_t}(\boldsymbol{s}_{t+1}), & \text{for } V\text{-function,} \\ Q_{\theta_t}(\boldsymbol{s}_t, \boldsymbol{a}_t) - \gamma Q_{\theta_t}(\boldsymbol{s}_{t+1}, \boldsymbol{a}_{t+1}), & \text{for } Q\text{-function,} \end{cases} \tag{4}$$

where $\boldsymbol{x}_t = \{\boldsymbol{s}_t, \boldsymbol{a}_t, \boldsymbol{s}_{t+1}, \boldsymbol{a}_{t+1}\}$, $V_{\theta_t}(\boldsymbol{s}_t) := (V_{\theta_t}(s_t^{(1)}), V_{\theta_t}(s_t^{(2)}), \ldots, V_{\theta_t}(s_t^{(n)}))^T$, and $Q_{\theta_t}(\boldsymbol{s}_t, \boldsymbol{a}_t) := (Q_{\theta_t}(s_t^{(1)}, a_t^{(1)}), Q_{\theta_t}(s_t^{(2)}, a_t^{(2)}), \ldots, Q_{\theta_t}(s_t^{(n)}, a_t^{(n)}))^T$. In this paper, we focus on Q-functions only, however, our algorithm also works for V-functions as indicated below. The KTD framework works under the Gaussian assumption, i.e., $\theta_t$ follows a Gaussian distribution at each stage $t = 1, 2, \ldots$. To address the nonlinearity of the function $h(\boldsymbol{x}, \theta)$, the KOVA algorithm, as proposed by Shashua & Mannor (2020), employs the Extended Kalman Filter (EKF) technique for calculating the mean and covariance matrices of $\theta_t$. This approach involves linearizing $h(\boldsymbol{x}, \theta)$ based on the first-order Taylor expansion, namely:

$$h(\boldsymbol{x}_t, \theta) \approx h(\boldsymbol{x}_t, \hat{\mu}_{t-1}) + \nabla_\theta h(\boldsymbol{x}_t, \hat{\mu}_{t-1})^T (\theta - \hat{\mu}_{t-1}),$$

where $\hat{\mu}_{t-1}$ denotes the estimator for the mean of $\theta_{t-1}$. The KOVA algorithm, detailed in Algorithm S1, however, encounters several significant challenges: (i) the approximation accuracy for the true filtering distribution of $\theta_t$ is unknown; (ii) it exhibits high computational complexity $O(np^2)$; and (iii) it demands considerable memory complexity, necessitating $O(p^2)$ additional space for the covariance matrix. Alternatively, Geist & Pietquin (2010) recommended the implementation of KTD using Unscented Kalman Filter (UKF). Nonetheless, this alternative algorithm encounters similar challenges to those faced by KOVA, including issues related to approximation accuracy, computational complexity, and memory requirements.

## 3 LANGEVINIZED KALMAN TEMPORAL DIFFERENCE ALGORITHM

To overcome the limitations encountered by existing KTD algorithms, we introduce an approach that integrates KTD with the LEnKF algorithm, leading to the development of the Langevinized Kalman Temporal Difference (LKTD) algorithm. The LEnKF is a reformulation of the Ensemble Kalman filter (EnKF) (Evensen, 1994) under the framework of Langevin dynamics. The LEnKF inherits the forecast-analysis procedure from the EnKF and the use of minibatch data from the Stochastic Gradient Langevin Dynamics (SGLD) algorithm (Welling & Teh, 2011), making it scalable with respect to the state dimension $p$ and the mini-batch size $n$. Distinctively, the LEnKF algorithm is designed to converge to the accurate filtering distribution, setting it apart from the traditional EnKF.

### 3.1 THE LKTD ALGORITHM

The LKTD algorithm is designed to solve the RL problem by framing it within the state-space model outlined in equation (2). As previously explained, this model differs from the model (1) by incorporating the prior information, $\pi(\theta)$, into the state evolution equation. This incorporation generally enhances the robustness of the algorithm, especially when using a deep neural network to approximate the function $h(\boldsymbol{x}, \theta)$. Next, we can apply the variance splitting technique (Zhang et al., 2023) to convert the model (1) into a state-space model with a linear measurement equation, while allowing the state evolution equation to be nonlinear. The variance splitting technique can be described as follows.

Without loss of generality, let's assume that $\eta_t \sim N(0, \sigma^2 I_n)$ for each stage $t$, where $I_n$ is an $n \times n$-identity matrix. By the state augmentation approach, we define

$$\varphi_t = \begin{pmatrix} \theta_t \\ \xi_t \end{pmatrix}, \quad \xi_t = h(\boldsymbol{x}_t; \theta_t) + u_t, \quad u_t \sim N(0, \alpha\sigma^2 I_n), \tag{5}$$

where $\xi_t$ is an $n$-dimensional vector, and $0 < \alpha < 1$ is a pre-specified constant. Suppose that $\theta_t$ has a prior distribution $\pi(\theta)$ as specified previously, the joint density function of $\varphi_t = (\theta_t^\top, \xi_t^\top)^\top$ is given by $\pi(\varphi_t) = \pi(\theta_t)\pi(\xi_t|\theta_t)$, where $\xi_t|\theta_t \sim N(h(\boldsymbol{x}_t; \theta_t), \alpha\sigma^2 I)$. Based on Langevin dynamics, we can reformulate (2) as the following model:

$$\begin{aligned} \varphi_t &= \varphi_{t-1} + \frac{\epsilon_t}{2}\frac{n}{\mathcal{N}}\nabla_\varphi \log \pi(\varphi_{t-1}) + \tilde{w}_t, \\ \boldsymbol{r}_t &= H_t\varphi_t + v_t, \end{aligned} \tag{6}$$

where $\mathcal{N} > 0$, $\tilde{w}_t \sim N(0, \frac{n}{\mathcal{N}}B_t)$, $B_t = \epsilon_t I_{\tilde{p}}$, $\tilde{p} = p + n$ is the dimension of $\varphi_t$; $H_t = (\boldsymbol{0}, I_n)$ such that $H_t\varphi_t = \xi_t$; $v_t \sim N(0, (1-\alpha)\sigma^2 I_n)$, which is independent of $\tilde{w}_t$ for all $t$. We call $\mathcal{N}$ the pseudo-population size, which scales uncertainty of the estimator of the system. Refer to Lemma 1 and Theorem 1 for mathematical justifications for this issue.

By (5) and (6), $\boldsymbol{x}_t$, $\boldsymbol{\theta}_t$, $\xi_t$ and $\boldsymbol{r}_t$ form a hierarchical model with the conditional distribution

$$\xi_t|\boldsymbol{r}_t, \boldsymbol{x}_t, \boldsymbol{\theta}_t \sim \mathcal{N}(\alpha\boldsymbol{r}_t + (1-\alpha)h(\boldsymbol{x}_t; \boldsymbol{\theta}_t), \alpha(1-\alpha)\sigma^2 I_n), \tag{7}$$

which, as shown by Zhang et al. (2023), eventually leads to an efficient particle filtering algorithm for handling the state-space models with a nonlinear measurement equation. Similar to LEnKF, we adopt the forecast-analysis procedure from EnKF to the model (6), leading to Algorithm 1. It works in a single chain, different from particle filtering algorithms. The time complexity of the algorithm is $O(np)$. This attractive time complexity is due to the special structure of $H_t$, rendering the matrix $K_t$ and equation 10 easily computed. Regarding the settings of $\mathcal{K}$ and $\alpha$, we make the following remarks: First, as indicated by the Kalman gain matrix $K_{t,k}$, only the $\xi$-component of $\varphi_{t,k}$ is updated at each analysis step. Generally, $\xi_{t,k}$ converges rapidly, benefiting from second-order gradient information. Thus, $\mathcal{K}$ does not need to be excessively large. Based on the property (7), Zhang et al. (2023) demonstrated that LEnKF acts as a variance reduction version of SGLD if $0.5 < \alpha < 1$, recommending $\alpha$ be set close to 1. In this paper, we default $\mathcal{K} = 5$ and $\alpha = 0.9$, initializing $\xi_{t,0}$ by $\boldsymbol{r}_t$ at each time step $t$ to enhance the convergence of the simulation.

### 3.2 CONVERGENCE THEORY

To study the convergence of Algorithm 1, it suffices to study the convergence of Algorithm S2, which ignores the inner iterations for imputing the latent variables $\xi_t$ and serves as the prototype of Algorithm 1. Lemma 1 shows that Algorithm S2 is actually an accelerated preconditioned SGLD algorithm. Its proof follows Theorem S1 of Zhang et al. (2023) .

**Lemma 1.** *Algorithm S2 implements a preconditioned SGLD algorithm, for which*

$$\varphi_t^a = \varphi_{t-1}^a + \frac{\epsilon_t}{2}\Sigma_t\nabla_\varphi \log \pi(\varphi_{t-1}^a|\boldsymbol{z}_t) + e_t, \tag{11}$$

*where $\boldsymbol{z}_t = (\boldsymbol{r}_t, \boldsymbol{x}_t)$ as defined in Algorithm S2, $\Sigma_t = \frac{n}{\mathcal{N}}(I - K_t H_t)$ is a constant matrix given $\varphi_t$, $e_t \sim N(0, \epsilon_t\Sigma_t)$, and the gradient term $\nabla_\varphi \log \pi(\varphi_{t-1}^a|\boldsymbol{z}_t)$ is given by $\nabla_\varphi \log \pi(\varphi_{t-1}^a|\boldsymbol{z}_t) = \frac{\mathcal{N}}{n}\sum_{i=1}^n \nabla_\varphi \log \pi(z_t^{(i)}|\varphi_{t-1}^a) + \nabla_\varphi \log \pi(\varphi_{t-1}^a)$.*

**Algorithm 1:** Langevinized Kalman Temporal-Difference (LKTD)

---

**Initialization:** Draw $\theta_0^a \in \mathbb{R}^p$ drawn from the prior distribution $\pi(\theta)$.

**for** $t=1,2,\ldots,T$ **do**

    **Sampling:** With policy $\rho_{\theta_{t-1}^a}$, generate a set of $n$ transition tuples, denoted by
$\boldsymbol{z}_t = (\boldsymbol{r}_t, \boldsymbol{x}_t) := \{r_t^{(j)}, x_t^{(j)}\}_{j=1}^n$, where $x_t^{(j)} = (s_t^{(j)}, a_t^{(j)}, s_{t+1}^{(j)}, a_{t+1}^{(j)})^T$ and
$x_t^{(j)} = (s_t^{(j)}, a_t^{(j)}, s_{t+1}^{(j)})^T$ correspond to the choices of the $Q$-function and $V$-function
in (4), respectively.

    **for** $k=1,2,\ldots,\mathcal{K}$ **do**

        **Presetting:** Set $B_{t,k} = \epsilon_{t,k} I_{\tilde{p}}$, $R_t = 2(1-\alpha)\sigma^2 I$, and the Kalman gain matrix
$K_{t,k} = B_{t,k} H_t^\top (H_t B_{t,k} H_t^\top + R_t)^{-1}$.

        **Forecast:** Draw $\tilde{w}_{t,k} \sim N_p(0, \frac{n}{\mathcal{N}} B_{t,k})$ and calculate

$$\varphi_{t,k}^f = \varphi_{t,k-1}^a + \frac{\epsilon_{t,k}}{2} \frac{n}{\mathcal{N}} \nabla_\varphi \log \pi(\varphi_{t,k-1}^a) + \tilde{w}_{t,k}, \tag{8}$$

        where $\varphi_{t,0}^a = ({\theta_{t-1,\mathcal{K}}^a}^\top, \boldsymbol{r}_t^\top)^\top$ if $k=1$, and the gradient term is given by

$$\nabla_\varphi \log \pi(\varphi_{t,k-1}^a) = \begin{pmatrix} \nabla_\theta \log \pi(\theta_{t,k-1}) + \frac{1}{\alpha\sigma^2} \frac{\mathcal{N}}{n} \nabla_\theta h(\boldsymbol{x}_t; \theta_{t,k-1})(\xi_{t,k-1} - h(\boldsymbol{x}_t; \theta_{t,k-1})) \\ -\frac{1}{\alpha\sigma^2}(\xi_{t,k-1} - h(\boldsymbol{x}_t; \theta_{t,k-1})) \end{pmatrix}.$$
$$\tag{9}$$

        **Analysis:** Draw $v_{t,k} \sim N_n(0, \frac{n}{\mathcal{N}} R_t)$ and calculate

$$\varphi_{t,k}^a = \varphi_{t,k}^f + K_{t,k}(\boldsymbol{r}_t - H_t \varphi_{t,k}^f - v_{t,k}) = \varphi_{t,k}^f + K_{t,k}(\boldsymbol{r}_t - \boldsymbol{r}_{t,k}^f). \tag{10}$$

    **end**

**end**

---

To establish the convergence of the preconditioned SGLD sampler (11), it suffices to establish the convergence of the conventional SGLD sampler in the context of reinforcement learning by noting the positive definiteness of the preconditioned matrix. Specifically, we have

$$\Sigma_t = \frac{n}{\mathcal{N}}(I - K_t H_t) = \frac{n}{\mathcal{N}}[I - \epsilon_t H^T (\epsilon_t H_t H_t^T + R_t)^{-1} H_t],$$

which implies $\Sigma_t$ has bounded positive eigenvalues for all $t \geq 1$.

### 3.2.1 Convergence of the LKTD algorithm under the on-policy setting

With a slight abuse of notations, we would prove the convergence of the following SGLD sampler in the RL context:

$$\theta_k = \theta_{k-1} + \epsilon_k G(\theta_{k-1}, \boldsymbol{z}_k) + \sqrt{2\beta^{-1}\epsilon_k}\,\mathfrak{e}_k, \tag{12}$$

where $\mathfrak{e}_k \sim N(0, I_d)$, $\beta$ is the inverse temperature, and $k$ indexes stages of the RL process. In pursuit of our objective, we introduce Assumption A1 as detailed in the Appendix. This assumption aligns with the conditions outlined in Raginsky et al. (2017) to demonstrate the convergence of SGLD in simulating a posterior distribution with a fixed dataset. To tailor the sampler for RL, a context where the total sample size can be considered infinitely large, we introduce the pseudo-population size $\mathcal{N}$ to prevent the degeneration issue of the invariant distribution of $\theta_k$, thereby reformulating RL as a sampling problem rather than an optimization problem. As a result, the proposed method can perform robustly with respect to the dynamics of the distribution $\pi(\boldsymbol{z}|\theta_k)$. However, under appropriate assumptions (see Remark 1), correct inference for the optimal policy can still be made based on the stationary distribution $\nu_{\mathcal{N}}(\theta) \propto \exp(-\beta\mathcal{G}(\theta))$, where $\mathcal{G}(\theta) = O(\mathcal{N})$ as stated in Theorem 1. Theorem 1 and the followed Corollary 1 establish the convergence of the LKTD algorithm under the general nonlinear setting for the value and Q-functions. Their proofs are given in Appendix.

**Theorem 1.** *Consider the SGLD sampler (12) with a polynomailly-decay learning rate $\epsilon_k = \frac{\epsilon_0}{k^\varpi}$ for some $\varpi \in (0,1)$. Suppose the environment is stationary and Assumption A1 holds. If $\mathbb{E}(G(\theta_{k-1}, z_k)) = g(\theta_{k-1})$ holds for any stage $k \in \{1,\ldots,K\}$, $\beta \geq 1 \vee \frac{2}{m_U}$, then*

*there exist constants $(C_0, C_1, C_2, C_3)$ independent of the learning rates such that for all $K \in \mathbb{N}$, the 2-Wasserstein distance between $\mu_K$ and $\nu_{\mathcal{N}}$ can be upper bounded by*

$$\mathcal{W}_2(\mu_K, \nu_{\mathcal{N}}) \leq (12 + C_2 \epsilon_0 (\frac{1}{1-\varpi} K^{1-\varpi}))^{\frac{1}{2}} \cdot [(C_1 \epsilon_0^2 (\frac{2\varpi}{2\varpi - 1}) + \delta C_0 (\frac{\epsilon_0}{1-\varpi} K^{1-\varpi}))^{\frac{1}{2}}$$
$$+ (C_1 \epsilon_0^2 (\frac{2\varpi}{2\varpi - 1}) + \delta C_0 (\frac{\epsilon_0}{1-\varpi} K^{1-\varpi}))^{\frac{1}{4}}] + C_3 \exp\left(-\frac{1}{\beta c_{LS}} (\frac{\epsilon_0}{1-\varpi} K^{1-\varpi})\right), \tag{13}$$

*where $\mu_K(\theta)$ denotes the probability law of $\theta_K$, $\nu_{\mathcal{N}}(\theta) \propto \exp(-\beta \mathcal{G}(\theta))$, $\mathcal{G}(\theta) = O(\mathcal{N})$ is the anti-derivative of $g(\theta)$, i.e., $\nabla_\theta \mathcal{G}(\theta) = g(\theta)$, and $c_{LS}$ denotes a logarithmic Sobolev constant satisfied by the $\nu_{\mathcal{N}}$. In addition, the constants $(C_0, C_1, C_2, C_3)$ are given by*

$$C_0 = L_U^2 (\kappa_0 + 2(1 \vee \frac{e}{m_U})(b + 2B^2 + \frac{d}{\beta})) + B^2,$$

$$C_1 = 6 L_U^2 (C_0 + \frac{d}{\beta}), \quad C_2 = \kappa_0 + 2b + 2d,$$

$$C_3 = \sqrt{2 c_{LS} (\log \|\nu_0\|_\infty + \frac{d}{2} \log \frac{3\pi}{m_U \beta} + \beta (\frac{M_U \kappa_0}{3} + B\sqrt{\kappa_0} + A + \frac{b}{2} \log 3))}.$$

Regarding statistical inference with the samples $\{\theta_k : k = 1, 2, \ldots, K\}$, we have the remark:

**Remark 1.** *Let $\phi(\theta))$ be a test function, which is bounded and differentiable. Suppose that the conditions of Theorem 1 hold and $(\nu_{\mathcal{N}}(\theta), \phi(\theta))$ satisfies the Laplace regularity condition as given in Theorem 2.3 of Sun et al. (2022). Then, by Lapalce approximation, we have*

$$\bar{\phi}_{\mathcal{N}}(\theta) = \frac{\int \phi(\theta) \exp(-\beta \mathcal{G}(\theta)) d\theta}{\int \exp(-\beta \mathcal{G}(\theta)) d\theta} = \phi(\theta^*) + O(\frac{r_n^4}{\mathcal{N}}), \quad as \ K \to \infty, \tag{14}$$

*where $\theta^*$ denotes the maximizer of $\nu_{\mathcal{N}}(\theta)$ and thus the maximizer of $\nu_\infty(\theta)$, and $r_n$ denotes the connectivity of the sparse DNN learned for approximating the value or Q-function. Therefore, when the number of total time steps $K$ becomes large and $\mathcal{N} \succ r_n^4$ holds, we can make inference for the policy using the Monte Carlo average $\hat{\phi} = [\sum_{k=1}^K \epsilon_k \phi(\theta_k)]/[\sum_{k=1}^K \epsilon_k]$, which, by letting $K \to \infty$, forms a consistent estimator for $\phi(\theta^*)$, the true value of $\phi(\cdot)$ at the optimal policy.*

**Remark 2.** *The choice of the pseudo population size $\mathcal{N}$ reflects our trade-off between optimization and sampling. It acts as a tempering factor for the system. As $\mathcal{N} \to \infty$, we have $\mathcal{G}(\theta) \to \infty$ and, consequently, the stationary distribution $\nu_{\mathcal{N}}(\theta)$ degenerates to a delta function centered at $\theta^*$, as defined in Remark 1.*

We note that the proof for the convergence of $\hat{\phi}$ toward the population mean $\bar{\phi}_{\mathcal{N}}(\theta)$ requires that the learning rate sequence satisfies Assumption A2. This condition is readily met by the polynomailly-decay learning rate sequence outlined in Theorem 1. Finally, we note that the conclusions of Theorem 1 remains valid for Algorithm 1. Therefore, Remark 1 and Remark 2 also hold for this algorithm.

**Corollary 1.** *The conclusions of Theorem 1 and Remark 1 remains valid for Algorithm 1.*

### 3.2.2 COOPERATION WITH REPLAY BUFFER

In Section 3.2.1, we demonstrated the $\mathcal{W}_2$-convergence of LKTD algorithm under an on-policy framework, where transition tuples $\boldsymbol{z}_t$ are determined by the preceding parameter $\theta_{t-1}$. In contrast, off-policy algorithms obtain $\boldsymbol{z}_t$ from replay buffers, a strategy that boosts data efficiency and is frequently adopted in Q-learning algorithms. This section is dedicated to establishing a convergence theory for the LKTD algorithm when it is integrated with a replay buffer. In practice, the replay buffer $\mathcal{B}$ stores transition $\{z_{t,j}\}_{j=1}^J$ drawn from the stationary distribution $\pi(z|\theta_{t-1})$ at each time step $t$. Given its finite capacity, the replay buffer retains only the transition tuples from the last $R$ time steps. That is, the replay buffer at time step $t$, denoted by $\mathcal{B}_t$, contains only the transition tuples generated from $\{\pi(z|\theta_\tau)\}_{\tau=t-R}^{t-1}$. In the population scope, we define the replay buffer at time step $t$, denoted by $\bar{\pi}(z|\boldsymbol{\theta}_{t-1}^R)$, as

a mixture of finite number of stationary distributions $\{\pi(z|\theta_{t-i})\}_{i=1}^R$. The replay buffer can be explicitly written as

$$\bar{\pi}(z|\boldsymbol{\theta}_{t-1}^R) = \frac{1}{R}\sum_{i=1}^R \pi(z|\theta_{t-i}), \qquad (15)$$

where $\boldsymbol{\theta}_{t-1}^R = \{\theta_{t-i}\}_{i=1}^R$ and $R \in \mathbb{N}$. Given the buffer's structure, we assume that the samples drawn from it are $R$-dependent. That is, $z_t$ and $z_{t'}$ are independent for all $z_t \in \mathcal{B}_t$, $z_{t'} \in \mathcal{B}_{t'}$ and $|t - t'| > R$. For some test function $\phi(\theta)$ of interest, we define the posterior average as: $\bar{\phi} = \int_\Theta \phi(\theta)\nu_\mathcal{N}(\theta)d\theta$, where $\nu_\mathcal{N}(\theta)$ is the target distribution as defined in Theorem 1. Let $\{\theta_t\}_{t=1}^T$ be the samples generated from LKTD algorithm, and the sample average $\hat{\phi}$ is as defined in Remark 1. In Theorem 2, we show that although the gradient is biased due to replay buffer, the bias and variance of $\hat{\phi}$ vanish asymptotically. In other words, we can employ replay buffer to improve data-efficiency without losing the asymptotic consistency.

**Theorem 2.** *Let $\{\theta_t\}_{t=1}^T$ be a sequence of updates generated from LKTD with replay buffer. At each time $t$, the transition tuple $z_t$ is sampled from the replay buffer $\bar{\pi}(z_t|\boldsymbol{\theta}_{t-1}^R)$. In addition to the assumptions in theorem 1, we further assume the following holds:*

*(i) (Lipschitz) $\int_\mathcal{Z} |\pi(z|\theta) - \pi(z|\vartheta)|^2 dz \le L\|\theta - \vartheta\|^2$;*

*(ii) (Integrability) $\int_\mathcal{Z} \|G(\theta, z)\|^2 dz \le M$ and $\int_\mathcal{Z} \|G(\theta, z)\|^2\pi(z|\theta)dz \le M$, $\forall \theta \in \Theta$.*

*Then for a bounded test function $\phi$, the bias of the LKTD can be bounded as:*

$$|\mathbb{E}\hat{\phi} - \bar{\phi}| = O(\frac{1}{S_T} + \frac{\sum_{t=1}^T \epsilon_t^2}{S_T}), \quad \mathbb{E}(\hat{\phi} - \bar{\phi})^2 = O(\frac{1}{S_T} + \frac{\sum_{t=1}^T \epsilon_t^2}{S_T^2} + \frac{(\sum_{t=1}^T \epsilon_t^2)^2}{S_T^2}) \qquad (16)$$

## 4    EXPERIMENTS

This section compares LKTD with prominent RL algorithms such as DQN, BootDQN (Osband et al., 2016), QR-DQN (Bellemare et al., 2017) and KOVA (Shashua & Mannor, 2020). Using an simple indoor escape environment, we demonstrate the advantages of LKTD in three aspects: (1) accuracy of Q-value estimation, (2) uncertainty quantification of Q-values, and (3) optimal policy exploration. Furthermore, by employing more complex environments like OpenAI gym, we demonstrate that LKTD is capable of learning better and more stable policies for both training and testing.

### 4.1    INDOOR ESCAPE ENVIRONMENT

Figure 1 depicts a simple indoor escape environment, for which the state space consists of 100 grids and the agent's objective is to navigate to the goal positioned at the top right corner. The agent starts its task from the bottom left grid at time $t = 0$. For every time step $t$, the agent identifies its current position, represented by the coordinate $s = (x, y)$. Based on a policy, the agent chooses an action $a \in \{N, E, S, W\}$. The action taken by the agent determines the adjacent grid to which it moves. Following each action, the agent is awarded an immediate reward, $r_t$, drawn from the distribution $\mathcal{N}(-1, 0.01)$. It's worth noting

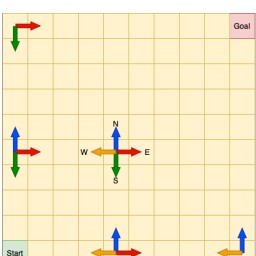

Figure 1: Indoor escape environment

that for most states, the Q-values for actions N and E are identical. This highlights the importance of exploring diverse optimal policies to achieve a consistent and resilient policy. Our experiment showcases the sampling framework are capable of learning a mixed optimal policy in a single run. We compare the proposed sampling framework LKTD against existing RL algorithms like DQN, BootDQN, QR-DQN and KOVA in the training of the deep Q-network. Refer to section A.4.2 for the detailed experimental setup.

For each algorithm, we collect the last 3000 parameter updates to form a $\theta$-sample pool, denoted by $\boldsymbol{\theta}_s = \{\hat{\theta}_i\}$, which naturally induces a sample pool of Q-functions $\mathbf{Q}_s = \{Q_\theta(\cdot, \cdot)|\theta \in \boldsymbol{\theta}_s\}$. We can obtain a point estimate of the Q-value at $(s, a)$ by calculating the sample average $\hat{Q}(s, a) = \frac{1}{n}\sum_{i=1}^n Q_{\hat{\theta}_i}(s, a)$. For uncertainty quantification, we can achieve one-step

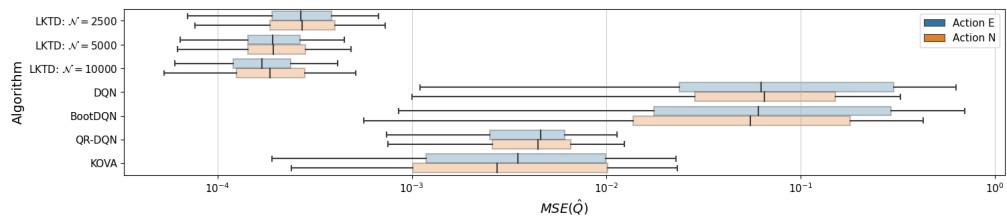

Figure 2: Boxplots for $\text{MSE}(\hat{Q}_a)$ (for $a \in \{N, E\})$)

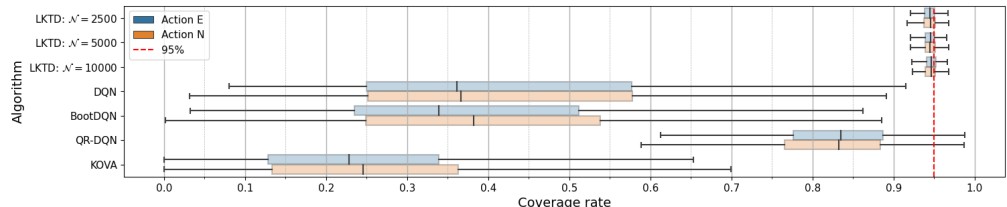

Figure 3: Boxplots for coverage rates (for $a \in \{N, E\}$))

value tracking by constructing a 95% prediction interval with the Q-value samples. Due to the simplicity of this environment, the Q-values of optimal policy, denoted by $Q^*(s, a)$, can be calculated by Monte Carlo simulations. For each algorithm and parameter setting, we conduct 100 runs and calculate two metrics: (1) the mean squared error (MSE) between $\hat{Q}(s, a)$ and $Q^*(s, a)$, denoted by $\text{MSE}(\hat{Q}_a)$ for each action $a$, where the average is taken over all states $s$; and (2) the coverage rate (CR) of the 95% prediction intervals.

Figure 2 presents a boxplot illustrating the distribution of $\text{MSE}(\hat{Q}_a)$ (for $a \in \{N, E\}$) across 100 experiments for each algorithm. Here, we consider only the actions $a \in \{N, E\}$, since $\{S, W\}$ are sub-optimal actions at all states and the corresponding Q-values cannot be well approximated due to the lack of enough transition tuples on them. Figure 2 indicates that LKTD yields notably higher Q-value estimation accuracy compared to all other algorithms. Moreover, the plot shows a clear trend that as the pseudo population size $\mathcal{N}$ grows, their accuracy correspondingly improves. For uncertainty quantification, Figure 3 shows that the coverage rates from the LKTD algorithm is close to the nominal 95% and independent of the choice of pseudo population size, whereas the DQN, BootDQN and KOVA algorithms fail to construct correct prediction intervals. Although QR-DQN approximates the distribution of the Q-function, it does not provide the correct interval estimation for Q-values. These observations on LKTD align well with the point we made in Remark 2.

Effective policy exploration is crucial for RL agents as it empowers them to adeptly learn various optimal policies and navigate challenges like the local-trap issue. In this specific environment, the Q-values for actions $N$ and $E$ are indistinguishable. Therefore, an algorithm excelling in policy exploration should, during training, give equal consideration to both optimal actions. To quantify policy exploration, we introduce the concept of **mean policy probability**. For a given policy pool $\varrho$ at state $s$, it's defined as:

$$p_\varrho(a|s) = \frac{1}{|\varrho|} \sum_{\rho \in \varrho} \mathbf{1}_a(\rho(s)),$$

where $\varrho$ represents the policy pool derived from the $\theta$-sample pool obtained in a run of the algorithm. In simpler terms, the mean policy probability measures the frequency of an action chosen by the policy sample at a specific state. Figure 4 shows that LKTD effectively explores the optimal policies across the majority of grids, whereas DQN sticks to one optimal policy, failing to explore others. This implies that DQN tends to be locally trapped in RL, compared to LKTD.

From a computational aspect, the LKTD algorithm stands out for its efficiency and scalability. As detailed in Table A3, we have recorded the average computation time required by each algorithm to execute a single parameter update. The findings indicate that LKTD scale effectively in relation to network and batch size. Their time complexities align closely with that of DQN. Conversely, the KOVA algorithm, due to its reliance on the calculation of the Jacobian matrix and matrix inversion, proves to be computationally less efficient.

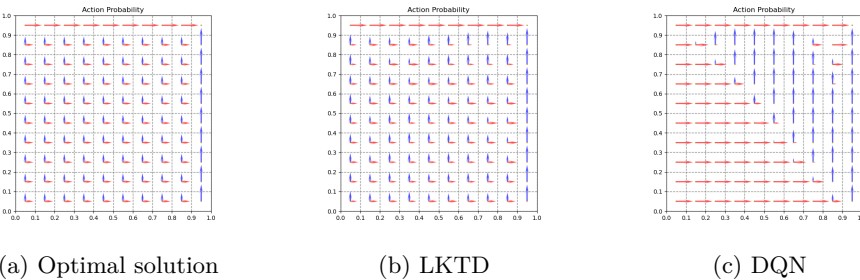

|                     |              |              |
|:-------------------:|:------------:|:------------:|
| (a) Optimal solution | (b) LKTD     | (c) DQN      |

Figure 4: Mean policy probabilities for the indoor escape environment: (a) known optimal solution; (b) learned by LKTD; (c) learned by DQN, failing to explore different policies.

## 4.2 CLASSICAL CONTROL PROBLEMS

This section evaluates LKTD's performance on four OpenAI gym challenges: Acrobot-v1, CartPole-v1, LunarLander-v2, and MountainCar-v0, comparing it against DQN and QR-DQN based on RL Baselines3 Zoo (Raffin, 2020) training framework. We conducted 100 replicates per experiment, with results in Figures 5 and A4, excluding the top and bottom 5% outliers. Mean reward curves are shown with solid lines, and a 90% confidence interval is indicated by the colored regions, showcasing LKTD's efficient exploration and robustness. For experimental details, see section A.5.

Across all tested environments, LKTD consistently surpasses DQN and slightly better than QR-DQN. LKTD's ability to achieve markedly higher training reward indicates its capacity to learn more robust policies against sub-optimal actions generated by random exploration. This superiority is particularly pronounced in sensitive environments like CartPole-v1 (see Figure 5), where even a single misstep can lead to episode termination. Even when discounting the noise from random exploration, LKTD's evaluation rewards remain superior to DQN's. Additionally, when it comes to optimal policy exploration, LKTD can identify a more optimal model within an equivalent computational timeframe as DQN, highlighting its efficiency in policy exploration.

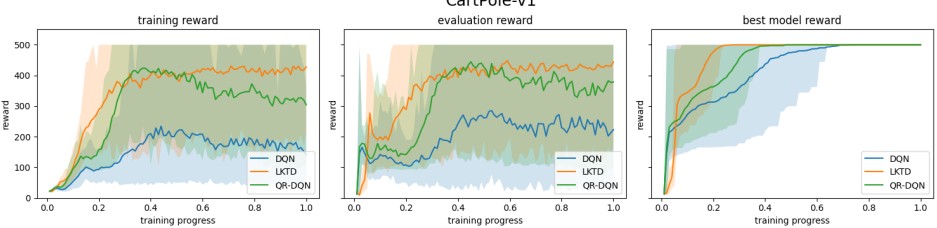

Figure 5: CartPole-v1: The left plot shows the cumulative rewards obtained during the training process, the middle plot shows the testing performance without random exploration, and the right plot shows the performance of best model learned up to the point $t$.

## 5 CONCLUSION

In this paper, we present a novel sampling framework designed to enhance SGD optimizers for addressing deep reinforcement learning challenges. By redefining the state-space model and introducing a pseudo population size, we enable SGMCMC algorithms, such as LKTD and SGLD, to converge to the accurate posterior distribution under mild conditions. Our approach outperforms existing value-based algorithms in benchmarks. Specifically, our LKTD algorithm demonstrates greater computational efficiency and circumvents potential matrix degeneration issues by eliminating the need for linearization, unlike the KOVA algorithm. Compared to DQN and its variants, our framework not only provides more precise point estimates of Q-values but also generates accurate prediction intervals for value tracking. In gym environments, LKTD surpasses DQN and QR-DQN in robustness and efficiency of optimal policy discovery. Overall, our framework signifies a significant advancement in deep RL optimization, offering improvements in both efficiency and precision.

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

# A  APPENDIX

## A.1  EXTENDED KALMAN TEMPORAL DIFFERENCE ALGORITHM

Let $\hat{\Sigma}_t$ denote the estimator for the covariance matrix of $\theta_t$. Additionally, for equation (1), we let $W_t \in \mathbb{R}^{p \times p}$ denote the covariance matrix of the Gaussian noise $w_t$, and let $\Gamma_t \in \mathbb{R}^{n \times n}$ denote the covariance matrix of the Gaussian noise $\eta_t$. The resulting KTD algorithm is Algorithm S1 given in the Appendix. The major issues with Algorithm S1 are (i) unknown approximation accuracy: it is unclear how well $N(\hat{\mu}_t, \hat{\Sigma}_t)$ approximates the true distribution of $\theta_t$; (ii) computational complexity: it is $O(np^2)$ per iteration; and (iii) memory complexity: it requires $O(p^2)$ additional space to store the covariance matrix $\hat{\Sigma}_t$.

---

**Algorithm S1:** Extended Kalman Temporal Difference Algorithm (KOVA Algorithm); Shashua & Mannor (2020)

---

**Initialize** $\hat{\mu}_0$;
**for** $t=1,2,\ldots,T$ **do**

    **(i)** Set predictions: $\hat{\mu} = \hat{\mu}_{t|t-1} = \hat{\mu}_{t-1}$ and $\hat{\Sigma}_{t|t-1} = \hat{\Sigma}_{t-1} + W_t$;
    **(ii)** Generate $n$ transition tuples $\{\boldsymbol{r}_t, h(\boldsymbol{x}_t, \hat{\mu})\}$ from the system via the
    agent-environment interaction;
    **(iii)** Calculate $(p \times n)$-dim matrix $\nabla_\theta h(\boldsymbol{x}_t, \hat{\mu})$ and the Kalman gain matrix

$$K_t = \hat{\Sigma}_{t|t-1} \nabla_\theta h(\boldsymbol{x}_t, \hat{\mu}) \Gamma_{\tilde{\boldsymbol{r}}_t}^{-1},$$

    where $\Gamma_{\tilde{\boldsymbol{r}}_t} = \nabla_\theta h(\boldsymbol{x}_t, \hat{\mu})^T \hat{\Sigma}_{t|t-1} \nabla_\theta h(\boldsymbol{x}_t, \hat{\mu}) + \Gamma_t$;
    **(iv)** Update the mean and covariance matrix estimators:

$$\hat{\mu}_t = \hat{\mu}_{t|t-1} + \tilde{\alpha} K_t (\boldsymbol{r}_t - h(\boldsymbol{x}_t, \hat{\mu}_{t|t-1})),$$
$$\hat{\Sigma}_t = \hat{\Sigma}_{t|t-1} - \tilde{\alpha} K_t \Gamma_{\tilde{\boldsymbol{r}}_t} K_t^T,$$

    where $\tilde{\alpha}$ is the learning rate.
**end**

---

## A.2  THE PROTOTYPE OF THE LKTD ALGORITHM

By ignoring the detail of state augmentation, the model (6) can be simulated using Algorithm S2, which is the single-chain version of Algorithm 2 of Zhang et al. (2023).

---

**Algorithm S2:** Prototype of the LKTD Algorithm

---

**Initialization:** Start with an initial parameter sample $\varphi_0^a \in \mathbb{R}^p$, drawn from the prior
  distribution $\pi(\varphi)$;
**for** $t=1,2,\ldots,T$ **do**
    **Presetting:** Set $B_t = \epsilon_t I_p$, $R_t = 2\sigma^2 I_n$, and the Kalman gain matrix
    $K_t = B_t H_t^\top (H_t B_t H_t^\top + R_t)^{-1}$;
    **Sampling:** With policy $\rho_{\varphi_{t-1}^a}$, generate a set of $n$ transition tuples from the
    stationary distribution $\mu_{\varphi_{t-1}^a}$, denoted by $\boldsymbol{z}_t = (\boldsymbol{r}_t, \boldsymbol{x}_t) = \{z_{t,j}\}_{j=1}^n$;
    **Forecast:** Draw $w_t \sim N_p(0, \frac{n}{\mathcal{N}} B_t)$ and calculate

$$\varphi_t^f = \varphi_{t-1}^a + \frac{\epsilon_t}{2} \frac{n}{\mathcal{N}} \nabla \log \pi(\varphi_{t-1}^a) + w_t. \tag{A1}$$

    **Analysis:** Draw $v_t \sim N_n(0, \frac{n}{\mathcal{N}} R_t)$ and calculate

$$\varphi_t^a = \varphi_t^f + K_t(\boldsymbol{r}_t - H_t \varphi_t^f - v_t) := \varphi_t^f + K_t(\boldsymbol{r}_t - \boldsymbol{r}_t^f). \tag{A2}$$

**end**

---

**Assumption A1.**

(C1) *For any $\theta \in \Theta$, we are able to generate tuples $\boldsymbol{z}$ from a unique stationary distribution $\pi(z|\theta)$, the function $G : \Theta \times \mathcal{Z}$ is measurable, and $\|g(\theta)\| = \|\int_{\mathcal{Z}} G(\theta, z)\pi(z|\theta)dz\| < \infty$.*

(C2) *There exists a function $\mathcal{G}(\theta)$, which is an anti-derivative of $g(\theta)$ with respect to $\theta$, i.e., $\nabla_\theta \mathcal{G}(\theta) = g(\theta)$, such that $|\mathcal{G}(0)| \leq A$ for some constant $A > 0$; in addition, there exists some constant $B > 0$ such that $\|g(0)\| \leq B$.*

(C3) *There exists some constant $L_U > 0$ such that*
$$\|g(\theta) - g(\vartheta)\| \leq L_U \|\theta - \vartheta\|, \quad \forall \theta, \vartheta \in \Theta.$$

(C4) *The function $\mathcal{G}(\theta)$ is $(m_U, b)$-dissipative; that is, for some $m_U > 0$ and $b \geq 0$,*
$$\langle \theta, g(\theta) \rangle \geq m_U \|\theta\|^2 - b, \quad \forall \theta \in \Theta.$$

(C5) *There exist a constant $\delta$ and some constants $M_U$ and $B$ such that*
$$\mathbb{E}\|G(\theta, z) - g(\theta)\|^2 \leq 2\delta(M_U^2 \|\theta\|^2 + B^2), \quad \forall \theta \in \Theta.$$
*where the expectation is taken with respect to $z \sim \pi(z|\theta)$.*

(C6) *The probability law $\mu_0$ of the initial hypothesis $\theta_0$ has a bounded and strictly positive density $p_0$ with respect to the Lebesgue measure on $\Theta$, and*
$$\kappa_0 := \log \int_\Theta e^{\|\theta\|^2} p_0(\theta)d\theta < \infty.$$

In particular, the condition (C4) is quite standard for establishing the existence of an invariant distribution for $\theta$, see e.g. Raginsky et al. (2017) and Xu et al. (2018a). It intuitively indicates that the dynamics stays inside a bounded domain in high probability; if $\theta_k$ is far away from the origin, then the dynamics are forced to get back around the origin.

**Assumption A2.** *The learning rate sequence $\{\epsilon_t\}$ is decreasing, i.e., $0 < \epsilon_{k+1} < \epsilon_k$, and satisfies that*
$$i) \quad \sum_{k=1}^\infty \epsilon_k = \infty; \quad ii) \lim_{K \to \infty} \frac{\sum_{k=1}^K \epsilon_k^2}{\sum_{k=1}^K \epsilon_k} = 0.$$

A.3.2  PROOF OF THEOREM 1

*Proof.* The proof of Theorem 1 follows from Theorem 2 of Zhang et al. (2020). For convenience, we use $t$ and $k$ to index the continuous-time and discrete-time respectively. Firstly, we consider the following SDE
$$d\theta_t = -g(\theta_t)dt + \sqrt{2}d\mathcal{W}_t, \tag{A3}$$
where $g(\theta) = \int_{\mathcal{Z}} G(\theta, z)\pi(z|\theta)dz$. Let $\nu_t$ denote the distribution of $\theta_t$, and the stationary distribution of (A3) is denoted by $\nu_\infty$.
$$\theta_{k+1} = \theta_k - \epsilon_{k+1}G(\theta_k, z_{k+1}) + \sqrt{2\epsilon_{k+1}\beta^{-1}}\mathfrak{e}_{k+1}. \tag{A4}$$

Further, let $\mu_k$ denote the distribution of $\theta_k$ and $S_k = \sum_{i=1}^k \epsilon_i$. Since
$$W_2(\mu_K, \nu_\infty) \leq W_2(\mu_K, \nu_{S_K}) + W_2(\nu_{S_K}, \nu_\infty), \tag{A5}$$

we need to bound these two terms respectively.

For the first term, $W_2(\mu_K, \nu_{S_K})$, our proof is based on the proof of Theorem 2 in Zhang et al. (2020) with some modifications on learning rates. By definition, if $z_{k+1}$ is sampled from the stationary distribution $\pi(z|\theta_k)$, then $G(\theta_k, z_{k+1})$ is an unbiased estimator of $g(\theta_k)$,

i.e., $\mathbb{E}[G(\theta_k, z_{k+1})|\mathcal{F}_k] = g(\theta_k)$, $\forall \theta_k \in \Theta \subset \mathbb{R}^d$. And we define $p(t)$ which will be used in the following proof:

$$p(t) = \{k \in \mathbb{Z}|S_k \le t < S_{k+1}\} \tag{A6}$$

Then we focus on the following continuous-time interpolation of $\theta_k$:

$$\underline{\theta}(t) = \theta_0 - \int_0^t G(\underline{\theta}(S_{p(s)}), z_{p(s)+1})ds + \sqrt{\frac{2}{\beta}} \int_0^t d\mathcal{W}_s^{(d)} \tag{A7}$$

where $G \equiv G_k$ for $t \in [S_k, S_{k+1})$. And for each $k$, $\underline{\theta}(S_k)$ and $\theta_k$ have the same probability law $\mu_k$. Since $\underline{\theta}(t)$ is not a Markov process, we define the following process which has the same one-time marginals as $\underline{\theta}(t)$

$$V(t) = \theta_0 - \int_0^t H_s(V(s))ds + \sqrt{\frac{2}{\beta}} \int_0^t d\mathcal{W}_s^{(d)} \tag{A8}$$

with

$$H_t(x) := \mathbb{E}\left[G(\underline{\theta}(S_{p(t)}), z_{p(t)+1})|\underline{\theta}(t) = x\right] \tag{A9}$$

Let $\mathbf{P}_V^t := \mathcal{L}(V(s) : 0 \le s \le t)$ and $\mathbf{P}_\theta^t := \mathcal{L}(\theta(s) : 0 \le s \le t)$ and according to the proof of Lemma 3.6 in Raginsky et al. (2017), we can derive a similar result for the relative entropy of $\mathbf{P}_V^t$ and $\mathbf{P}_\theta^t$:

$$
\begin{aligned}
D_{KL}(\mathbf{P}_V^t||\mathbf{P}_\theta^t) &= -\int d\mathbf{P}_V^t \log \frac{d\mathbf{P}_\theta^t}{d\mathbf{P}_V^t} \\
&= \frac{\beta}{4} \int_0^t \mathbb{E}\|g(V(s)) - H_s(V(s))\|^2 ds \\
&= \frac{\beta}{4} \int_0^t \mathbb{E}\|g(\underline{\theta}(s)) - H_s(\underline{\theta}(s))\|^2 ds
\end{aligned}
\tag{A10}
$$

The last line follows the fact that $\mathcal{L}(\underline{\theta}(s)) = \mathcal{L}(V(s))$, $\forall s$. Then we will let $t = \sum_{k=1}^K \epsilon_k$ and we can use the martingale property of the integral to derive:

$$
\begin{aligned}
D_{KL}(\mathbf{P}_V^{\sum_{k=1}^K \epsilon_k}||\mathbf{P}_\theta^{\sum_{k=1}^K \epsilon_k}) &= \frac{\beta}{4} \sum_{j=0}^{K-1} \int_{S_j}^{S_{j+1}} \mathbb{E}\|g(\underline{\theta}(s)) - H_s(\underline{\theta}(s))\|^2 ds \\
&= \frac{\beta}{2} \sum_{j=0}^{K-1} \int_{S_j}^{S_{j+1}} \mathbb{E}\|g(\underline{\theta}(s)) - g(\underline{\theta}(S_j))\|^2 ds \\
&\quad + \frac{\beta}{2} \sum_{j=0}^{K-1} \int_{S_j}^{S_{j+1}} \mathbb{E}\|g(\underline{\theta}(S_j)) - H_s(\underline{\theta}(S_j))\|^2 ds \\
&= \frac{\beta L_U^2}{2} \sum_{j=0}^{K-1} \int_{S_j}^{S_{j+1}} \mathbb{E}\|\underline{\theta}(s) - \underline{\theta}(S_j)\|^2 ds \\
&\quad + \frac{\beta}{2} \sum_{j=0}^{K-1} \int_{S_j}^{S_{j+1}} \mathbb{E}\|g(\underline{\theta}(S_j)) - H_s(\underline{\theta}(S_j))\|^2 ds
\end{aligned}
$$

$$\tag{A11}$$
$$\tag{A12}$$

For the first part (A11), we consider some $s \in [S_j, S_{j+1})$, for which the following holds:

$$
\begin{aligned}
\underline{\theta}(s) - \underline{\theta}(S_j) &= -(s - S_j)G(\theta_k, z_{k+1}) + \sqrt{\frac{2}{\beta}}(\mathcal{W}_s^{(d)} - \mathcal{W}_{S_j}^{(d)}) \\
&= -(s - S_j)g(\theta_k) + (s - S_j)(g(\theta_k) - G(\theta_k, z_{k+1})) + \sqrt{\frac{2}{\beta}}(\mathcal{W}_s^{(d)} - \mathcal{W}_{S_j}^{(d)})
\end{aligned}
\tag{A13}
$$

Thus, we can use Lemma 3.1 and 3.2 in Raginsky et al. (2017) for the following result:

$$
\begin{aligned}
\mathbb{E}\|\underline{\theta}(s) - \underline{\theta}(S_j)\|^2 &\le 3\epsilon_{j+1}^2 \mathbb{E}\|g(\theta_j)\|^2 + 3\epsilon_{j+1}^2 \mathbb{E}\|g(\theta_j) - G(\theta_j, z_{j+1})\|^2 + 6\epsilon_{j+1}d \\
&\le 12\epsilon_{j+1}^2(L_U^2 \mathbb{E}\|\theta_j\|^2 + B^2) + \frac{6\epsilon_{j+1}d}{\beta}
\end{aligned}
\tag{A14}
$$

Hence we can bound the first part, (choosing $\epsilon_0 \leq 1$),

$$\frac{L_U^2}{2}\sum_{j=0}^{K-1}\int_{S_j}^{S_{j+1}}\mathbb{E}\|\underline{\theta}(s)-\underline{\theta}(S_j)\|^2 ds \leq \frac{L_U^2}{2}\sum_{j=0}^{K-1}[12\epsilon_{j+1}^3(L_U^2\mathbb{E}\|\theta_j\|^2+B^2)+\frac{6\epsilon_{j+1}^2 d}{\beta}]$$

$$\leq L_U^2(\sum_{j=0}^{K-1}\epsilon_{j+1}^2)\max_{0\leq j\leq K-1}[6(L_U^2\mathbb{E}\|\theta_j\|^2+B^2)+\frac{3d}{\beta}]$$

(A15)

$$\leq L_U^2(\frac{\pi^2}{6}\epsilon_0^2)\max_{0\leq j\leq K-1}[6(L_U^2\mathbb{E}\|\theta_j\|^2+B^2)+\frac{3d}{\beta}] \quad \text{(A16)}$$

The second part (A12) can be bounded as follows:

$$\frac{1}{2}\sum_{j=0}^{K-1}\int_{S_j}^{S_{j+1}}\mathbb{E}\|g(\underline{\theta}(S_j))-H_s(\underline{\theta}(S_j))\|^2 ds = \frac{1}{2}\sum_{j=0}^{K-1}\epsilon_{j+1}\mathbb{E}\|g(\theta_j)-G(\theta_j,z_{j+1})\|^2$$

$$\leq \delta S_K \max_{0\leq j\leq K-1}(L_U^2\mathbb{E}\|\theta_j\|^2+B^2)$$

$$\leq \delta\epsilon_0(1+\log(K))\max_{0\leq j\leq K-1}(L_U^2\mathbb{E}\|\theta_j\|^2+B^2)$$

Due to the data-processing inequality for the relative entropy, we have

$$D_{KL}(\mu_K\|\nu_{S_K}) \leq D_{KL}(\mathbf{P}_V^t\|\mathbf{P}_\theta^t)$$

$$\leq \frac{L_U^2}{2}\sum_{j=0}^{K-1}\int_{S_j}^{S_{j+1}}\mathbb{E}\|\underline{\theta}(s)-\underline{\theta}(S_j)\|^2 ds + \frac{1}{2}\sum_{j=0}^{K-1}\int_{S_j}^{S_{j+1}}\mathbb{E}\|g(\underline{\theta}(S_j))-H_s(\underline{\theta}(S_j))\|^2 ds$$

$$\leq L_U^2(\sum_{j=0}^{K-1}\epsilon_{j+1}^2)\max_{0\leq j\leq K-1}[6(L_U^2\mathbb{E}\|\theta_j\|^2+B^2)+\frac{3d}{\beta}]+\delta S_K\max_{0\leq j\leq K-1}(L_U^2\mathbb{E}\|\theta_j\|^2+B^2)$$

$$\leq L_U^2\epsilon_0^2(\frac{2\varpi}{2\varpi-1})\max_{0\leq j\leq K-1}[6(L_U^2\mathbb{E}\|\theta_j\|^2+B^2)+\frac{3d}{\beta}]$$

$$+\delta\epsilon_0(\frac{1}{1-\varpi}K^{1-\varpi})\max_{0\leq j\leq K-1}(L_U^2\mathbb{E}\|\theta_j\|^2+B^2)$$

According to the proof of Lemma 3.2 in Raginsky et al. (2017), we can bound the term $\mathbb{E}\|\theta_k\|^2$

$$\mathbb{E}\|\theta_{k+1}\|^2 \leq (1-2\epsilon_{k+1}m_U+4\epsilon_{k+1}^2 M_U^2)\mathbb{E}\|\theta_k\|^2+2\epsilon_{k+1}b+4\epsilon_{k+1}^2 B^2+\frac{2\epsilon_{k+1}d}{\beta}$$

Similar to the statement of Lemma 3.2 in Raginsky et al. (2017), we can fix $\epsilon_0 \in (0, 1 \wedge \frac{m_U}{4M_U^2} \wedge \frac{1}{m_U})$. Then, we can know that

$$\mathbb{E}\|\theta_{k+1}\|^2 \leq (1-\epsilon_{k+1}m_U)\mathbb{E}\|\theta_k\|^2+2\epsilon_{k+1}(b+2B^2+\frac{d}{\beta}) \qquad \text{(A17)}$$

where $\epsilon_K$ is the minimum of the decreasing learning rate sequence. There are two cases to consider.

- If $1-2\epsilon_K m_U+4\epsilon_K^2 M_U^2 \leq 0$, then from (A17) it follows that

$$\mathbb{E}\|\theta_{k+1}\|^2 \leq 2\epsilon_0(b+B^2+\frac{d}{\beta})$$

$$\leq \mathbb{E}\|\theta_0\|^2+2(b+B^2+\frac{d}{\beta})$$

- If $0 \leq 1 - 2\epsilon_K m_U + 4\epsilon_K^2 M_U^2 \leq 1$, then iterating (A17) gives

$$
\mathbb{E}\|\theta_k\|^2 \leq (1 - \epsilon_k m_U)\mathbb{E}\|\theta_{k-1}\|^2 + 2\epsilon_k(b + 2B^2 + \frac{d}{\beta})
$$

$$
\leq e^{-\epsilon_k m_U}\mathbb{E}\|\theta_{k-1}\|^2 + 2\epsilon_k(b + 2B^2 + \frac{d}{\beta})
$$

$$
\leq e^{-m_U S_k}\mathbb{E}\|\theta_0\|^2 + 2(b + 2B^2 + \frac{d}{\beta})\sum_{i=1}^{k}\epsilon_i e^{-m_U(S_k - S_i)}
$$

$$
\leq \mathbb{E}\|\theta_0\|^2 + 2(b + 2B^2 + \frac{d}{\beta})e^{-m_U S_k}\sum_{i=1}^{k}\epsilon_i e^{m_U S_i}
$$

$$
\leq \mathbb{E}\|\theta_0\|^2 + 2(b + 2B^2 + \frac{d}{\beta})e^{-m_U S_k} \cdot e^{m_U \epsilon_0}\int_0^{S_k}e^{m_U x}dx
$$

$$
\leq \mathbb{E}\|\theta_0\|^2 + 2(b + 2B^2 + \frac{d}{\beta})e^{-m_U S_k} \cdot e^{m_U \epsilon_0}(\frac{1}{m_U}e^{m_U S_k} - \frac{1}{m_U})
$$

$$
\leq \mathbb{E}\|\theta_0\|^2 + 2(b + 2B^2 + \frac{d}{\beta})\frac{e^{m_U \epsilon_0}}{m_U}
$$

$$
\leq \mathbb{E}\|\theta_0\|^2 + 2(b + 2B^2 + \frac{d}{\beta})\frac{e}{m_U}
$$

Now, we have

$$
\max_{0 \leq j \leq K-1}(L_U^2 \mathbb{E}\|\theta_j\|^2 + B^2) \leq (L_U^2(\kappa_0 + 2(1 \vee \frac{e}{m_U})(b + 2B^2 + \frac{d}{\beta})) + B^2) := C_0
$$

We denote the $6L_U^2(C_0 + \frac{d}{\beta})$ as $C_1$ and we can derive

$$
D_{KL}(\mu_K\|\nu_{S_K}) \leq C_1\epsilon_0^2(\frac{2\varpi}{2\varpi - 1}) + \delta C_0\epsilon_0(\frac{1}{1 - \varpi}K^{1-\varpi})
$$

Then according to Lemma 3.3 in Raginsky et al. (2017), if we denote $\kappa_0 + 2b + 2d$ as $C_2$, we can derive the following result:

$$
W_2(\mu_K, \nu_{S_K}) \leq (12 + C_2 S_K)^{\frac{1}{2}} \cdot [D_{KL}(\mu_K\|\nu_{S_K})^{\frac{1}{2}} + D_{KL}(\mu_K\|\nu_{S_K})^{\frac{1}{4}}]
$$

$$
\leq (12 + C_2\epsilon_0(\frac{1}{1 - \varpi}K^{1-\varpi}))^{\frac{1}{2}} \cdot [(C_1\epsilon_0^2(\frac{2\varpi}{2\varpi - 1}) + \delta C_0\epsilon_0(\frac{1}{1 - \varpi}K^{1-\varpi}))^{\frac{1}{2}}
$$

$$
+ (C_1\epsilon_0^2(\frac{2\varpi}{2\varpi - 1}) + \delta C_0\epsilon_0(\frac{1}{1 - \varpi}K^{1-\varpi}))^{\frac{1}{4}}]
$$

Now we derive the bound for $W_2(\nu_{S_K}, \nu_\infty)$. By following the results in Raginsky et al. (2017) that there exist some positive constants $(C_3, c_{LS})$,

$$
W_2(\nu_{S_K}, \nu_\infty) \leq C_3 \exp\left(-\frac{S_K}{c_{LS}}\right)
$$

$\square$

### A.3.3 Proof of Corollary 1

*Proof.* By Theorem 1 of Ma et al. (2015), Algorithm 1 works as a pre-conditioned SGLD algorithm with the pre-conditioner $\Sigma_t$, and it has the same stationary distribution as the SGLD algorithm (12). By (13), we have the 2-Wasserstein distance convergence for algorithm (12) under the given assumptions. Therefore, for Algorithm 1, we also have $\mathcal{W}_2(\mu_T, \nu_{\mathcal{N}}) \to 0$ as $T \to \infty$ by noting that $\Sigma_t$ is positive definite for any $t$. $\square$

*Proof.* By following the proof in Chen et al. (2015), we define the functional $\psi$ that solves the Poisson Equation:

$$\mathcal{L}\psi(\theta_t) = \phi(\theta_t) - \bar{\phi} \tag{A18}$$

And $\psi$ satisfies the following smoothness condition

**Assumption A3.** *$\psi$ and its up to 3rd-order derivatives, $\mathcal{D}^k\psi$, are bounded by a function $\mathcal{V}$, i.e., $\|\mathcal{D}^k\psi\| \leq C_k\mathcal{V}^{p_k}$ for $k = (0,1,2,3)$, $C_k$, $p_k > 0$. Furthermore, the expectation of $\mathcal{V}$ on $\{\theta_t\}$ is bounded: $\sup_{s\in(0,1)} \mathcal{V}^p(s\theta + (1-s)Y) \leq C(\mathcal{V}^p(\theta) + \mathcal{V}^p(Y))$, $\forall \theta$, $Y$, $p \leq \max\{2p_k\}$ for some $C > 0$.*

**Assumption A4.** *Let $\Sigma_t$ be the preconditioner, and assume that $\lambda_{t,\ell} \leq \inf_k \lambda_{\min}(\Sigma_t) \leq \sup_k \lambda_{\max}(\Sigma_t) \leq \lambda_{t,u}$ for some $\lambda_{t,\ell}$ and $\lambda_{t,u}$, where $\lambda_{\max}(\cdot)$ and $\lambda_{\min}(\cdot)$ denote the largest and smallest eigenvalues, respectively.*

First let us denote

$$\tilde{\mathcal{L}}_t = \Sigma_t G(\theta_{t-1}, z_t) \cdot \nabla_\theta + \frac{1}{2}\Sigma_t\Sigma_t^\top : \nabla_\theta\nabla_\theta^\top \tag{A19}$$

the local generator of LKTD with replay buffer, where $\boldsymbol{a} \cdot \boldsymbol{b} := \boldsymbol{a}^\top\boldsymbol{b}$ is the vector inner product, $\boldsymbol{A} : \boldsymbol{B} := \text{tr}\{\boldsymbol{A}^\top\boldsymbol{B}\}$ is the matrix double dot product. Furthermore, let $\mathcal{L}$ be the true generator of the LKTD without replay buffer, that is, replacing the stochastic gradient in $\tilde{\mathcal{L}}_t$ with the true gradient. As a result, we have the relation:

$$\tilde{\mathcal{L}}_t = \mathcal{L} + \Delta V_t, \tag{A20}$$

where $\Delta V_t := (G(\theta_{t-1}, z_t) - g(\theta_{t-1}))^\top \Sigma_t \nabla_\theta$, where $g(\theta_{t-1}) = \int_\mathcal{Z} G(\theta_{t-1}, z)\pi(z|\theta_{t-1})dz$ is the true gradient, and $G(\theta_{t-1}, z_t)$ is the stochastic gradient calculated using transition tuples sampled from the replay memory. By following the proof of Theorem 1 in Li et al. (2016), we can derive the estimation error as follows:

$$\hat{\phi} - \bar{\phi} = \frac{\mathbb{E}\psi(\theta_t) - \psi(\theta_0)}{S_T} + \frac{1}{S_T}\sum_{t=1}^{T-1}(\mathbb{E}\psi(\theta_{t-1}) - \psi(\theta_{t-1})) + \sum_{t=1}^{T}\frac{\epsilon_t}{S_T}\Delta V_t\psi(\theta_{t-1}) + O(\frac{\sum_{t=1}^T \epsilon_t^2}{S_T}) \tag{A21}$$

By taking expectation on both side, we derived the bias as:

$$|\mathbb{E}\hat{\phi} - \bar{\phi}| = O(\frac{1}{S_T} + \frac{\sum_{t=1}^T \epsilon_t\|\mathbb{E}\Delta V_t\|}{S_T} + \frac{\sum_{t=1}^T \epsilon_t^2}{S_T}) \tag{A22}$$

To prove the consistency of $\hat{\phi}$, we need to bound the term $\frac{1}{S_T}\sum_{t=1}^T \epsilon_t\|\mathbb{E}\Delta V_t\|$. By Assumption A3 and A4, it is sufficient to prove that $\sum_{t=1}^T \epsilon_t\|\mathbb{E}\zeta_t\|$ is bounded, where $\zeta_t = G(\theta_{t-1}, z_t) - g(\theta_{t-1})$ is the gradient bias at time $t$. Let $\bar{g}(\boldsymbol{\theta}_{t-1}^R) := \int_\mathcal{Z} G(\theta_{t-1}, z)\bar{\pi}(z|\boldsymbol{\theta}_{t-1}^R)dz$ be the expectation of the biased gradient given the replay buffer $\bar{\pi}(z|\boldsymbol{\theta}_{t-1}^R)$. By applying assumption (i) and (ii), we can bound the conditional expectation of the gradient bias

$$\begin{aligned}
\|\mathbb{E}[\zeta_t|\mathcal{F}_{t-1}]\|^2 &= \|\bar{g}(\boldsymbol{\theta}_{t-1}^R) - g(\theta_{t-1})\|^2 \\
&\leq \|\int_\mathcal{Z} G(\theta_{t-1}, z) \cdot (\bar{\pi}(z|\boldsymbol{\theta}_{t-1}^R) - \pi(z|\theta_{t-1}))dz\|^2 \\
&\leq (\int_\mathcal{Z} \|G(\theta_{t-1}, z)\| \cdot |\bar{\pi}(z|\boldsymbol{\theta}_{t-1}^R) - \pi(z|\theta_{t-1})|dz)^2 \\
&\leq \int_\mathcal{Z} \|G(\theta_{t-1}, z)\|^2 dz \cdot \int_\mathcal{Z} |\bar{\pi}(z|\boldsymbol{\theta}_{t-1}^R) - \pi(z|\theta_{t-1})|^2 dz \qquad \text{(A23)} \\
&\leq M\int_\mathcal{Z} \frac{1}{R}\sum_{i=1}^R |\pi(z|\theta_{t-i}) - \pi(z|\theta_{t-1})|^2 dz \\
&\leq \frac{ML}{R}\sum_{i=1}^R \|\theta_{t-i} - \theta_t\|^2
\end{aligned}$$

By Jensen's inequality,

$$\|\mathbb{E}[\mathbb{E}[\zeta_t|\mathcal{F}_{t-1}]]\|^2 \leq \mathbb{E}\|\mathbb{E}[\zeta_t|\mathcal{F}_{t-1}]\|^2$$

$$\leq \mathbb{E}[\frac{ML}{R}\sum_{i=1}^{R}\|\theta_{t-i} - \theta_t\|^2]$$

$$\leq C\frac{1}{R}\sum_{i=1}^{R}\mathbb{E}\|\theta_{t-i} - \theta_{t-1}\|^2 \tag{A24}$$

$$\leq C\frac{1}{R}\sum_{i=1}^{R}\sum_{j=1}^{i-1}(i-1)\mathbb{E}\|\theta_{t-j} - \theta_{t-j-1}\|^2$$

Now, we want to bound the expected square difference between parameter updates

$$\mathbb{E}\|\theta_t - \theta_{t-1}\|^2 = \mathbb{E}\|\epsilon_t\Sigma_t G(\theta_{t-1}, z_t) + \sqrt{2\epsilon_t}e_t\|^2$$

$$\leq 2\epsilon_t^2\|\Sigma_t\|^2\mathbb{E}\|G(\theta_{t-1}, z_t)\|^2 + 4\epsilon_t\|\Sigma_t\|$$

$$\leq 2\epsilon_t^2\lambda_{\max}^2\mathbb{E}\|G(\theta_{t-1}, z_t)\|^2 + 4\epsilon_t\lambda_{\max} \tag{A25}$$

$$\leq O(\epsilon_t^2)$$

Combine the results in (A24) and (A25), we have

$$\|\mathbb{E}[\zeta_t]\|^2 \leq C\frac{1}{R}\sum_{i=1}^{R}\sum_{j=1}^{i-1}(i-1)\mathbb{E}\|\theta_{t-j} - \theta_{t-j-1}\|^2$$

$$\leq C\frac{1}{R}\sum_{i=1}^{R}\sum_{j=1}^{i-1}(i-1)O(\epsilon_{t-j}^2) \tag{A26}$$

$$\leq C\frac{R-1}{2}O(\epsilon_{t-R}^2)$$

$$\leq O(\epsilon_{t-R}^2) = O(\epsilon_t^2)$$

By (A26), the operator norm $\|\mathbb{E}\Delta V_t\|$ is of order $O(\epsilon_t)$. Combining this result with (A22) we can derive the bound for the bias

$$|\mathbb{E}\hat{\phi} - \bar{\phi}| = O(\frac{1}{S_T} + \frac{\sum_{t=1}^{T}\epsilon_t\|\mathbb{E}\Delta V_t\|}{S_T} + \frac{\sum_{t=1}^{T}\epsilon_t^2}{S_T})$$

$$= O(\frac{1}{S_T} + \frac{\sum_{t=1}^{T}\epsilon_t^2}{S_T} + \frac{\sum_{t=1}^{T}\epsilon_t^2}{S_T}) \tag{A27}$$

$$= O(\frac{1}{S_T} + \frac{\sum_{t=1}^{T}\epsilon_t^2}{S_T})$$

Now, consider the $L^2$ convergence of $\hat{\phi}$. Since $\mathbb{E}\Delta V_t$ is nonzero under the setting of replay buffer, we follow the proof of Theorem 3 in Chen et al. (2015) with some modification. The MSE of $\hat{\phi}$ can be written as

$$\mathbb{E}(\hat{\phi} - \bar{\phi})^2 \leq C\mathbb{E}\Big\{\frac{(\mathbb{E}\psi(\theta_t) - \psi(\theta_0))^2}{S_T^2} + \frac{\sum_{t=1}^{T-1}(\mathbb{E}\psi(\theta_{t-1}) - \psi(\theta_{t-1}))^2}{S_T^2} + \big(\sum_{t=1}^{T}\frac{\epsilon_t}{S_T}\Delta V_t\psi(\theta_{t-1})\big)^2$$

$$+ C(\frac{\sum_{t=1}^{T}\epsilon_t^2}{S_T})^2\Big\} \tag{A28}$$

Let $\Delta V_t = \mathbb{E}\Delta V_t + \delta_t^\top\Sigma_t\nabla_\theta$, where $\delta_t = (G(\theta_{t-1}, z_t) - \bar{g}(\boldsymbol{\theta}_{t-1}^R))$ has mean 0. Since $z_t$'s are $R$-dependent due to the structure of replay buffer, $Cov(\delta_t, \delta_{t'}) = 0$ for all $|t - t'| > R$. By assumption (ii), $\mathbb{E}\|\delta_t\|^2$ is bounded. Hence, we can derive the following bound:

$$\mathbb{E}\|\sum_{t=1}^{T}\frac{\epsilon_t}{S_T}\Delta V_t\|^2 \le 2\|\sum_{t=1}^{T}\frac{\epsilon_t}{S_T}\mathbb{E}\Delta V_t\|^2 + 2\mathbb{E}\|\sum_{t=1}^{T}\frac{\epsilon_t}{S_T}\delta_t\Sigma_t\nabla_\theta\|^2$$

$$\le 2(\sum_{t=1}^{T}\frac{\epsilon_t^2}{S_T^2})(\sum_{t=1}^{T}\|\mathbb{E}\Delta V_t\|^2) + 2C\sum_{|t-t'|<R}\frac{\epsilon_t\epsilon_{t'}}{S_T^2}Cov(\delta_t,\delta_{t'}) \tag{A29}$$

$$= O(\frac{(\sum_{t=1}^{T}\epsilon_t^2)^2}{S_T^2} + \frac{R\sum_{t=1}^{T}\epsilon_t^2\mathbb{E}\|\delta_t\|^2}{S_T^2})$$

$$= O(\frac{(\sum_{t=1}^{T}\epsilon_t^2)^2}{S_T^2} + \frac{\sum_{t=1}^{T}\epsilon_t^2}{S_T^2})$$

Finally, we can derive the MSE as

$$\mathbb{E}(\hat{\phi}-\bar{\phi})^2 = O(\frac{(\sum_{t=1}^{T}\epsilon_t^2)^2}{S_T^2} + \frac{\sum_{t=1}^{T}\epsilon_t^2}{S_T^2} + \frac{1}{S_T}) \tag{A30}$$

$\square$

## A.4 MORE NUMERICAL RESULTS

### A.4.1 IMPLEMENTATION OF THE SGLD AND SGHMC ALGORITHM

The notion of pseudo population introduced in the proposed LKTD algorithm can also be applied to SGLD and SGHMC algorithm. As implied by Lemma 1, we can directly implement equation (11) with $\Sigma_t$ being restricted to an identity matrix, which leaves the same stationary distribution. The pseudocode of SGLD and SGHMC are given in Algorithm S3 and Algorithm S4, where $\mathcal{K}$ is set to match the computation of LKTD.

---

**Algorithm S3:** SGLD for RL sampling framework

---

**Initialization:** Draw $\theta_0^a \in \mathbb{R}^p$ drawn from the prior distribution $\pi(\theta)$.

**for** $t=1,2,\ldots,T$ **do**

    **Sampling:** With policy $\rho_{\theta_{t-1}^a}$, generate a set of $n$ transition tuples, denoted by

    $\boldsymbol{z}_t = (\boldsymbol{r}_t, \boldsymbol{x}_t) := \{r_t^{(j)}, x_t^{(j)}\}_{j=1}^n$, where $x_t^{(j)} = (s_t^{(j)}, a_t^{(j)}, s_{t+1}^{(j)}, a_{t+1}^{(j)})^T$ and

    $x_t^{(j)} = (s_t^{(j)}, a_t^{(j)}, s_{t+1}^{(j)})^T$ correspond to the choices of the $Q$-function and $V$-function

    in (4), respectively.

    **for** $k=1,2,\ldots,\mathcal{K}$ **do**

        **Presetting:** Set $B_{t,k} = \epsilon_{t,k}I_{\tilde{p}}$.

        **Draw** $\tilde{w}_{t,k} \sim N_p(0, \frac{n}{\mathcal{N}}B_{t,k})$ and calculate

$$\theta_{t,k} = \theta_{t,k-1} + \frac{\epsilon_{t,k}}{2}\frac{n}{\mathcal{N}}\nabla_\theta\log\pi(\theta_{t,k-1}|\boldsymbol{z}_t) + \tilde{w}_{t,k}, \tag{A31}$$

        where $\theta_{t,0} = \theta_{t-1,\mathcal{K}}$ if $k = 1$, and the gradient term is given by

$$\nabla_\theta\log\pi(\theta_{t,k-1}|\boldsymbol{z}_t) = \nabla_\theta\log\pi(\theta_{t,k-1}) + \frac{1}{\sigma^2}\frac{\mathcal{N}}{n}\nabla_\theta h(\boldsymbol{x}_t;\theta_{t,k-1})(\boldsymbol{r}_t - h(\boldsymbol{x}_t;\theta_{t,k-1}))$$
$$\tag{A32}$$

    **end**

**end**

---

### A.4.2 INDOOR ESCAPING ENVIRONMENT

This section serves as a complement to Section 4.1 in the main text, offering more comprehensive experiment settings and numerical results to compare SGMCMC sampling algorithms with non-sampling algorithms. The SGMCMC sampling algorithms considered comprise

---

**Algorithm S4:** SGHMC for RL sampling framework

---

**Initialization:** Draw $\theta_0^a \in \mathbb{R}^p$ drawn from the prior distribution $\pi(\theta)$, momentum coefficient $\alpha$.

**for** $t=1,2,\ldots,\ T$ **do**

    **Sampling:** With policy $\rho_{\theta_{t-1}^a}$, generate a set of $n$ transition tuples, denoted by

    $\boldsymbol{z}_t = (\boldsymbol{r}_t, \boldsymbol{x}_t) := \{r_t^{(j)}, x_t^{(j)}\}_{j=1}^n$, where $x_t^{(j)} = (s_t^{(j)}, a_t^{(j)}, s_{t+1}^{(j)}, a_{t+1}^{(j)})^T$ and

    $x_t^{(j)} = (s_t^{(j)}, a_t^{(j)}, s_{t+1}^{(j)})^T$ correspond to the choices of the $Q$-function and $V$-function in (4), respectively.

    Set $v_{t,0} = 0$

    **for** $k=1,2,\ldots,\mathcal{K}$ **do**

        **Presetting:** Set $B_{t,k} = \epsilon_{t,k} I_{\tilde{p}}$.

        **Draw** $\tilde{w}_{t,k} \sim N_p(0, \alpha \frac{n}{\mathcal{N}} B_{t,k})$ and calculate

$$v_{t,k} = (1-\alpha)v_{t,k-1} + \frac{\epsilon_{t,k}}{2} \frac{n}{\mathcal{N}} \nabla_\theta \log \pi(\theta_{t,k-1}|\boldsymbol{z}_t) + \tilde{w}_{t,k}$$

$$\theta_{t,k} = \theta_{t,k-1} + v_{t,k}$$

        (A33)

        where $\theta_{t,0} = \theta_{t-1,\mathcal{K}}$ if $k=1$, and the gradient term is given by

$$\nabla_\theta \log \pi(\theta_{t,k-1}|\boldsymbol{z}_t) = \nabla_\theta \log \pi(\theta_{t,k-1}) + \frac{1}{\sigma^2} \frac{\mathcal{N}}{n} \nabla_\theta h(\boldsymbol{x}_t; \theta_{t,k-1})(\boldsymbol{r}_t - h(\boldsymbol{x}_t; \theta_{t,k-1}))$$

        (A34)

    **end**

**end**

---

LKTD, SGLD, and SGHMC, while the non-sampling algorithms encompass DQN, BootDQN, QR-DQN, and KOVA.

In this experiment, the Q-function is approximated by a deep neural network with two hidden layers of sizes (32, 32). The agent updates the network parameters every 10 interactions, for a total of $10^6$ action steps. The replay buffer size is set to $10^4$. For action selection, we use $\epsilon$-greedy exploration with a final exploring rate of $\epsilon = 0.01$. The batch size is 100. To achieve sparse deep neural network, we follow the suggestion in Sun et al. (2022), let the deep neural network parameters be subject to a mixture Gaussian prior:

$$\theta \sim (1-\lambda)\mathcal{N}(0,\sigma_0^2) + \lambda\mathcal{N}(0,\sigma_1^2) \qquad (A35)$$

where $\lambda \in (0,1)$ is the mixture proportion and $\sigma_0^2$ is usually set to a small number compare to $\sigma_1^2$. We set $\sigma_1 = 0.5$, $\sigma_0 = 0.05$ and $\lambda = 0.5$ in all SGMCMC simulations. In equation (2), the reward $r_t$ is assumed to be a Gaussian distribution with variance $\sigma^2$. For indoor escape environment, the reward is given by $\mathcal{N}(-1, 0.01)$; that is, we set $\sigma^2 = 0.01$.

For BootDQN, the number of "heads" is configured to 10, with the Bernoulli probability set at 0.5. For QR-DQN, the return distribution is approximated by 10 quantiles.

For this problem, the optimal policy is not unique, any policy that choose either action N or action E at any inner state is an optimal policy. Despite that there are multiple optimal policies, they all share the same Q-function, denoted by $Q^*(\cdot, \cdot)$. Since we adopt $\epsilon$-greedy exploration, we re-denoted $Q^*(\cdot, \cdot)$ by $Q_\epsilon^*(\cdot, \cdot)$ to indicates its dependence on the $\epsilon$-greedy exploration strategy. For all state-action pairs $(s, a)$, the Q-value $Q_\epsilon^*(s, a)$ can be estimated by Monte Carlo simulations. Note that $Q_\epsilon^*(\cdot, \cdot)$ is the target function that the deep neural network is to approximate.

For each algorithm, we collect from the last 3000 parameter updates to form a $\theta$-sample pool, denoted by $\boldsymbol{\theta}_s = \{\hat{\theta}_i\}$, which naturally induces a sample pool of Q-functions $\mathbf{Q}_s = \{Q_\theta(\cdot, \cdot)|\theta \in \boldsymbol{\theta}_s\}$. We can obtain a point estimate of the Q-value at $(s, a)$ by calculating the sample average $\hat{Q}(s, a) = \frac{1}{n} \sum_{i=1}^n Q_{\hat{\theta}_i}(s, a)$. For uncertainty quantification, we can achieve one-step value tracking by constructing a 95% prediction interval with the Q-value sample pool.

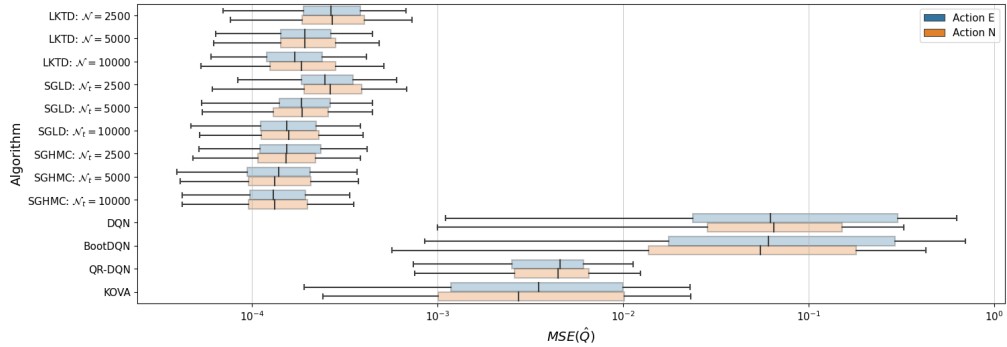

Figure A1: Boxplots for $\mathrm{MSE}(\hat{Q}_a)$ (for $a \in \{N, E\}$))

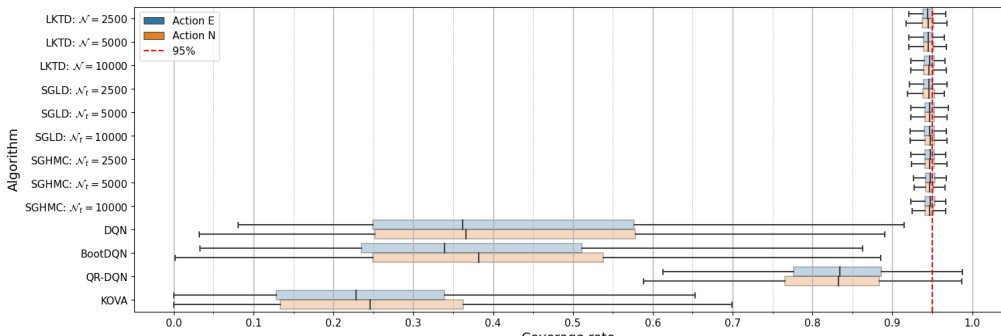

Figure A2: Boxplots for coverage rates

For each algorithm and parameter setting, we conduct 100 runs and calculate two metrics for each action at each run: (i) the mean squared error (MSE) between $\hat{Q}(s, a)$ and $Q_\epsilon^*(s, a)$, denoted by $\mathrm{MSE}(\hat{Q}_a)$, where the average is taken over all grids, that is,

$$\mathrm{MSE}(\hat{Q}_a) = \frac{1}{|\mathcal{S}|} \sum_{s \in \mathcal{S}} |\hat{Q}(s, a) - Q_\epsilon^*(s, a)|^2,$$

and (ii) the coverage rate (CR) of the 95% prediction interval of $Q_\epsilon^*(s, a)$, that is, the probability of $Q_\epsilon^*(s, a)$ falling inside the prediction interval. Figure A1 and Figure A2 show the boxplots of $\mathrm{MSE}(\hat{Q}_a)$ (with $a \in \{N, E\}$) and coverage rates, respectively. In Figure A1, the SGMCMC algorithms exhibit significantly smaller MSEs compared to other algorithms. It is worth noting that as the pseudo population increases, the MSE decreases, which supports the theoretical result in Remark 1. As shown in Figure A2, the coverage rates of all SGMCMC algorithms achieve the nominal 95% and independent of the choice of pseudo population size, whereas the DQN, BootDQN and KOVA algorithms fail to construct correct prediction intervals. Although the QR-DQN algorithm achieves a slightly higher coverage rate than DQN, the results prove that it does not converge to the correct return distribution.

In Table A1, we have recorded the trimmed mean (standard deviation) of $\mathrm{MSE}(\hat{Q}_a)$ (for $a \in \{N, E\}$) over 100 runs, where trimmed means are calculated by excluding the outliers. The outliers are the values that falls outside the interval (Q1-1.5IQR, Q3+1.5IQR), where Q1 and Q3 are, resepctively, the 1st and 3rd quartiles of the samples, and IQR = Q3 - Q1. Both tables indicate that SGMCMC algorithms are more accurate than non-sampling algorithms in Q-function approximation. Regarding uncertainty quantification, Table A2 presents the trimmed mean of the coverage rates and lengths of prediction intervals. It is worth noting that the prediction interval shrinks as the pseudo population size increases, which aligns with our theory as mentioned in Remark 1.

From a computational aspect, the LKTD and SGLD algorithms stand out for their efficiency and scalability, compared to the existing tracking algorithm KOVA. As detailed in Table A3, we have recorded the average computation time required by each algorithm to execute a

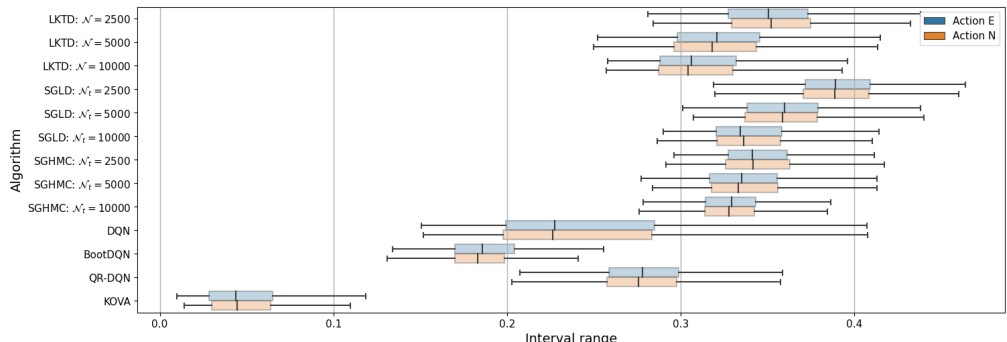

Figure A3: Boxplots for the range of prediction intervals

Table A1: Trimmed mean of $\mathrm{MSE}(\hat{Q}_a)$ $(a \in \{N, E\})$ over 100 runs

| Algorithm | $\epsilon_t$ | $\mathcal{N}$ | East | North |
|---|---|---|---|---|
| LKTD | 1e-5 | 2500 | 0.00031 (0.00018) | 0.00035 (0.00026) |
| LKTD | 1e-5 | 5000 | 0.00023 (0.00015) | 0.00023 (0.00015) |
| **LKTD** | **1e-5** | **10000** | **0.00020 (0.00012)** | **0.00022 (0.00013)** |
| SGLD | 1e-5 | 2500 | 0.00029 (0.00016) | 0.00031 (0.00016) |
| SGLD | 1e-5 | 5000 | 0.00021 (0.00010) | 0.00021 (0.00013) |
| **SGLD** | **1e-5** | **10000** | **0.00019 (0.00012)** | **0.00018 (0.00010)** |
| SGHMC | 1e-5 | 2500 | 0.00020 (0.00014) | 0.00021 (0.00021) |
| SGHMC | 1e-5 | 5000 | 0.00016 (0.00009) | 0.00017 (0.00011) |
| **SGHMC** | **1e-5** | **10000** | **0.00016 (0.00010)** | **0.00016 (0.00010)** |
| DQN | 1e-3 | - | 0.10890 (0.14942) | 0.08630 (0.10870) |
| BootDQN | 1e-3 | - | 0.11200 (0.18036) | 0.08757 (0.14150) |
| QR-DQN | 1e-2 | - | 0.00635 (0.00453) | 0.00583 (0.00342) |
| **KOVA** | **1** | **-** | **0.00584 (0.00812)** | **0.00533 (0.00771)** |

Table A2: Trimmed means of coverage rates and prediction interval widths over 100 runs

| Description | | | East | | North | |
|---|---|---|---|---|---|---|
| Algorithm | $\epsilon_t$ | $\mathcal{N}$ | CR | Range | CR | Range |
| **LKTD** | **1e-5** | **2500** | **0.94419 (0.00946)** | **0.35163 (0.03169)** | **0.94394 (0.01004)** | **0.35271 (0.03257)** |
| **LKTD** | **1e-5** | **5000** | **0.94458 (0.00952)** | **0.32217 (0.03195)** | **0.94440 (0.00989)** | **0.32079 (0.03223)** |
| **LKTD** | **1e-5** | **10000** | **0.94577 (0.01003)** | **0.31259 (0.03316)** | **0.94537 (0.00979)** | **0.31198 (0.03327)** |
| **SGLD** | **1e-5** | **2500** | **0.94411 (0.01316)** | **0.38984 (0.02707)** | **0.94474 (0.01141)** | **0.38938 (0.02712)** |
| **SGLD** | **1e-5** | **5000** | **0.94589 (0.01191)** | **0.35993 (0.02893)** | **0.94618 (0.00924)** | **0.35930 (0.02813)** |
| **SGLD** | **1e-5** | **10000** | **0.94636 (0.00878)** | **0.33496 (0.02148)** | **0.94648 (0.00906)** | **0.33559 (0.02167)** |
| **SGHMC** | **1e-5** | **2500** | **0.94633 (0.00890)** | **0.34193 (0.02125)** | **0.94578 (0.00931)** | **0.34144 (0.02140)** |
| **SGHMC** | **1e-5** | **5000** | **0.94704 (0.00854)** | **0.33553 (0.02439)** | **0.94622 (0.00885)** | **0.33586 (0.02458)** |
| **SGHMC** | **1e-5** | **10000** | **0.94682 (0.00893)** | **0.32756 (0.02000)** | **0.94659 (0.00874)** | **0.32659 (0.01990)** |
| DQN | 1e-3 | - | 0.41132 (0.20317) | 0.23791 (0.05742) | 0.41142 (0.19289) | 0.23736 (0.06317) |
| BootDQN | 1e-3 | - | 0.37995 (0.18066) | 0.19207 (0.04146) | 0.39634 (0.19053) | 0.18339 (0.02263) |
| **QR-DQN** | **1e-2** | **-** | **0.85690 (0.07660)** | **0.40063 (0.05326)** | **0.86395 (0.06097)** | **0.39800 (0.05206)** |
| KOVA | 1 | - | 0.24133 (0.15194) | 0.04756 (0.02534) | 0.25709 (0.15602) | 0.04987 (0.02886) |

single parameter update, utilizing an 4-core AMD Epyc 7662 Rome processor. The findings indicate that both LKTD and SGLD scale effectively in relation to network and batch size. Their time complexities align closely with that of DQN. Conversely, the KOVA algorithm, due to its reliance on the calculation of the Jacobian matrix and matrix inversion, proves to be computationally less efficient.

Table A3: Computation time for the indoor escaping example

| Algorithm | hidden layer | batch size | gradient steps (iterations) | cpu time ($\times 10^{-3}$) | time per iteration |
|-----------|-------------|-----------|----------------------------|-----------------------------|-------------------|
| LKTD | [32, 32] | 100 | 5 | 6.63 | 1.326 |
| LKTD | [32, 32] | 200 | 5 | 7.36 | 1.472 |
| LKTD | [64, 64] | 100 | 5 | 7.43 | 1.486 |
| SGLD | [32, 32] | 100 | 5 | 7.15 | 1.430 |
| SGLD | [32, 32] | 200 | 5 | 7.44 | 1.488 |
| SGLD | [64, 64] | 100 | 5 | 7.36 | 1.472 |
| SGHMC | [32, 32] | 100 | 5 | 7.47 | 1.494 |
| SGHMC | [32, 32] | 200 | 5 | 8.25 | 1.650 |
| SGHMC | [64, 64] | 100 | 5 | 8.08 | 1.616 |
| DQN | [32, 32] | 100 | 1 | 1.80 | 1.80 |
| DQN | [32, 32] | 200 | 1 | 2.32 | 2.32 |
| DQN | [64, 64] | 100 | 1 | 1.86 | 1.86 |
| BootDQN | [32, 32] | 100 | 1 | 2.29 | 2.29 |
| BootDQN | [32, 32] | 200 | 1 | 2.68 | 2.68 |
| BootDQN | [64, 64] | 100 | 1 | 2.26 | 2.26 |
| QR-DQN | [32, 32] | 100 | 1 | 2.41 | 2.41 |
| QR-DQN | [32, 32] | 200 | 1 | 2.95 | 2.95 |
| QR-DQN | [64, 64] | 100 | 1 | 2.51 | 2.51 |
| KOVA | [32, 32] | 100 | 1 | 44.20 | 44.20 |
| KOVA | [32, 32] | 200 | 1 | 87.00 | 87.00 |
| KOVA | [64, 64] | 100 | 1 | 251.00 | 251.00 |

A.5  CLASSIC CONTROL PROBLEMS

This section evaluates the performance of LKTD on four classical control problems in OpenAI gym (Brockman et al., 2016), including Acrobot-v1, CartPole-v1, LunarLander-v2 and MountainCar-v0. We compare LKTD with DQN and QR-DQN under the framework of RL Baselines3 Zoo (Raffin, 2020). The detailed hyperparameter setting is listed in Table A4 and Table A5. Each experiment is duplicated 100 times, and the training progress is recorded in Figure 5 and Figure A4. At each time step, the best and the worst 5% of the rewards are considered as outliers and excluded in the plots. Due to the adaptability of our sampling framework, LKTD can be easily applied to DQN algorithm by modifying the state-space model in equation (2) as

$$\theta_t = \theta_{t-1} + \frac{\epsilon_t}{2}\nabla \log \pi(\theta_{t-1}) + w_t,$$
$$\boldsymbol{y}_t = h(\boldsymbol{x}_t, \theta_t) + \eta_t,$$
(A36)

where $h(\boldsymbol{x}_t, \theta_t) = [Q_{\theta_t}(s_{t,1}, a_{t,1}), \ldots, Q_{\theta_t}(s_{t,n}, a_{t,n})]$ and $\boldsymbol{y}_t = \boldsymbol{r}_t + \gamma Q_{\theta_t}(\boldsymbol{s}_{t+1}, \boldsymbol{a}_{t+1})$. With suitable constraints on the semi-gradient, we can modify Theorem 1 to guarantee the convergence. In the four classic control problems, LKTD shows its strength in efficient exploration and robustness. In Figure A4, the lines represent the mean reward curves. For each algorithm, the colored area covers 90% of the reward curves. We consider 3 types of reward measurements, training reward, evaluation reward and the best model reward. Training reward records the cumulative reward during training, which include the $\epsilon$-exploration errors. Evaluation reward calculates the mean reward over 5 testing trails at 100 time point throughout the training progress. The best evaluation reward records the performance of the best learned model.

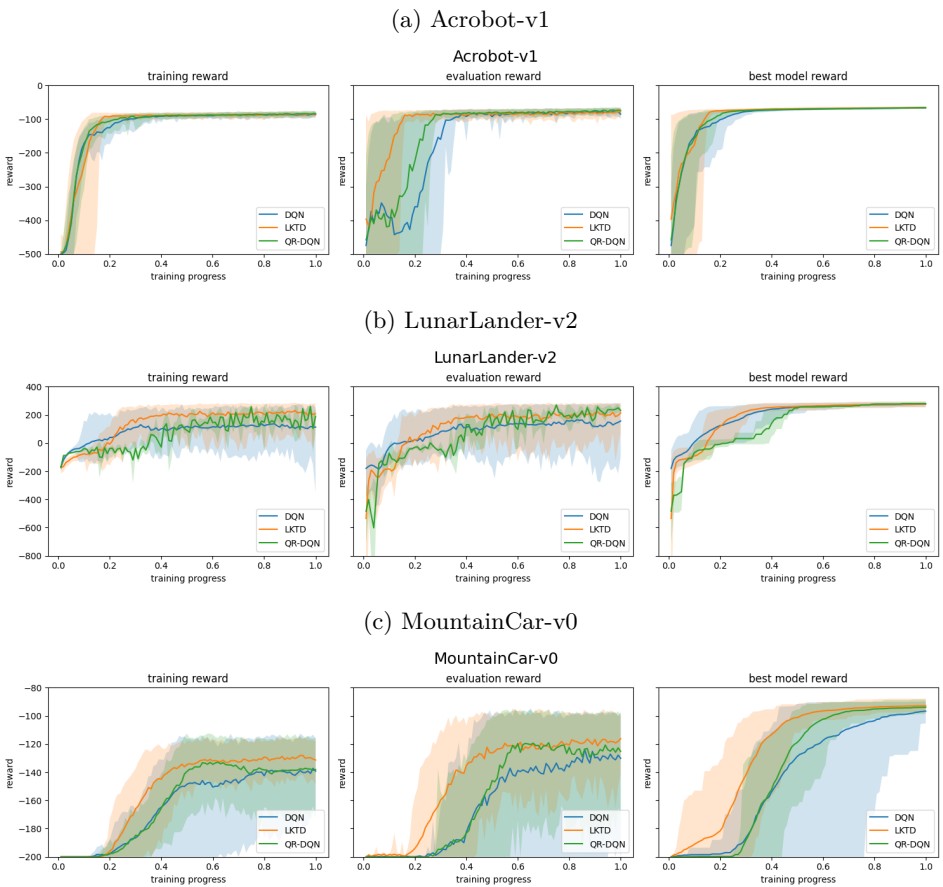

(a) Acrobot-v1

(b) LunarLander-v2

(c) MountainCar-v0

Figure A4: The first column shows the cumulative rewards obtained during the training process, the second column shows the testing performance without random exploration, and the third column shows the performance of best model learned up to the point $t$.

Table A4: Hyperparameters

| Environment | CartPole-v1 | | | MountainCar-v0 | | |
|---|---|---|---|---|---|---|
| Hyperparameters | LKTD | DQN | QR-DQN | LKTD | DQN | QR-DQN |
| learning rate | 2.5e-5 | 2.3e-3 | 2.3e-3 | 1.0e-4 | 4.0e-3 | 4.0e-3 |
| $\mathcal{N}$ (pseudo population) | 20000 | - | - | 20000 | - | - |
| $\sigma_\theta$ (prior) | 1 | - | - | 0.5 | - | - |
| $\sigma$ (observation) | 1 | - | - | 1 | - | - |
| target update interval | 1 | 10 | 10 | 100 | 600 | 600 |
| $\gamma$(discount factor) | 0.99 | 0.99 | 0.99 | 0.98 | 0.98 | 0.98 |
| training steps | 1e5 | 1e5 | 1e5 | 2e5 | 2e5 | 2e5 |
| batch size | 64 | 64 | 64 | 128 | 128 | 128 |
| buffer size | 1e4 | 1e5 | 1e5 | 1e4 | 1e5 | 1e5 |
| learning starts | 1000 | 1000 | 1000 | 0 | 0 | 0 |
| train freq | 4 | 256 | 256 | 32 | 16 | 16 |
| gradient steps | 1 | 128 | 128 | 16 | 8 | 8 |
| exploration fraction | 0.16 | 0.16 | 0.16 | 0.2 | 0.2 | 0.2 |
| exploration final eps | 0.04 | 0.04 | 0.04 | 0.07 | 0.07 | 0.07 |

Table A5: Hyperparameters (cont.)

| Environment | LunarLander-v2 | | | Acrobot-v1 | | |
|---|---|---|---|---|---|---|
| Hyperparameters | LKTD | DQN | QR-DQN | LKTD | DQN | QR-DQN |
| learning rate | 5.0e-6 | 6.3e-4 | 1.5e-3 | 5.0e-5 | 6.3e-4 | 6.3e-4 |
| $\mathcal{N}$ (pseudo population) | 20000 | - | - | 20000 | - | - |
| $\sigma_\theta$ (prior) | 1 | - | - | 1 | - | - |
| $\sigma$ (observation) | 1 | - | - | 1 | - | - |
| target update interval | 1 | 250 | 1 | 1 | 250 | 250 |
| $\gamma$(discount factor) | 0.99 | 0.99 | 0.995 | 0.99 | 0.99 | 0.99 |
| training steps | 2e5 | 2e5 | 2e5 | 1e5 | 1e5 | 1e5 |
| batch size | 128 | 128 | 128 | 128 | 128 | 128 |
| buffer size | 2.5e4 | 5e4 | 1e5 | 5e4 | 5e4 | 5e4 |
| learning starts | 0 | 0 | 10000 | 0 | 0 | 50000 |
| train freq | 4 | 4 | 256 | 4 | 4 | 4 |
| gradient steps | 4 | 4 | 256 | 4 | 4 | 4 |
| exploration fraction | 0.24 | 0.12 | 0.24 | 0.12 | 0.12 | 0.12 |
| exploration final eps | 0.05 | 0.10 | 0.18 | 0.05 | 0.1 | 0.1 |

