# OpenReview forum: "Fast Value Tracking for Deep Reinforcement Learning"
_ICLR.cc/2024/Conference — ICLR 2024 poster_

### Official Review · Reviewer_n3CS · 2023-11-01

**Soundness:** 2 fair
**Presentation:** 1 poor
**Contribution:** 3 good
**Rating:** 5
**Confidence:** 3

**Summary:**

This paper proposed a novel and scalable sampling algorithm for reinforcement learning. It provides a systematic theoretical analysis of the convergence, identifying that the method is able to quantify uncertainties.

**Strengths:**

This paper focuses on an important problem of RL: how to model the uncertainty during the interaction with the environment.

The paper proposed a new sampling framework for RL. The paper also gives a systematic theoretical analysis of the convergence of the proposed algorithm under the general nonlinear setting.

**Weaknesses:**

The writing of the paper is quite unclear to me. Some presentations need to be further clarified.

- I don't quite understand why the title of the paper is "fast value tracking". The term "tracking" is only mentioned in the introduction without clearly explaining it. Why is it "fast"?

- eq. 2,  introducing  $\pi$ seems lack motivation to me. Why it can tackle the issues suffered by KTD.



Some terms need further explanation and clarification.

- Please clarify what "stage" means.
- $x_t$ denotes a set of states and actions, does it refer to a trajectory containing a set of states and actions?
- It's weird to introduce a notation without explanation the first time, such as $h(\cdot)$
- "Optimal policy exploration"

The experiments are not sufficient

- the paper only compares the Adam algorithm.
- The improvement over Adam is not significant   according to Figure A1




Minor comments:

- It's kind of weird to say $r(s,a)$ is a random variable.
- Markov chain => Markov Chain
- It's kind of hard to get the main contribution of the paper from the introduction
- such as e.g.?
- The caption in Figure 5 is very akward.

**Questions:**

The paper claims that:

> it is more appropriate to consider the value or the model parameters as random variables rather than fixed unknowns, and it is preferable to focus on tracking rather than convergence during the policy learning process.

However, could you please explain what tracking is? And how could (i)(ii)(iii)(iv) can lead to this conclusion or statement?





How's the paper related to some random value function-based method? such as [1].

[1] Osband, Ian, et al. "Deep Exploration via Randomized Value Functions." *J. Mach. Learn. Res.* 20.124 (2019): 1-62.

---

> ### Author Response · Authors · 2023-11-23
> **Reply to Reviewer n3CS**
>
> ### reply to W1:
> We use the term "value tracking" to reflect the state-space model nature of reinforcement learning (RL). Specifically, we reformulate RL as a state space model and estimate the model using a modified Kalman filter method along with process of data generation.
>
> #### Ans:
>
> The word `fast' reflects the efficiency of the proposed LKTD algorithm,
>          As analyzed at the end of Section 1, LKTD has a computational complexity of $O(np)$ per iteration, allowing scalability for large-scale neural network approximations, where $p$ denotes the dimension (i.e., number of parameters) of the neural network. In contrast, the computational complexity of the existing filter algorithms is $O(p^2)$ per iteration.
>          Therefore, the proposed algorithm can be very efficient when a large-scale network is used in approximation, while the sample size at each stage/step is small.
>
> #### Ans:
> KTD suffers from inaccurate linearization and resulted operations on large-scale matrices. The proposed algorithm employs a state augmentation approach, see equation (5) of the paper, to avoid linearization. Furthermore, it accelerates computation  using the forecast-analysis procedure developed in ensemble Kalman filter, which avoids direct decomposition of the preconditioned matrix given in equation (11) of the paper. Refer to Zhang et al. [1] for more discussions on this issue.
>
> [1] Peiyi Zhang, Qifan Song, and Faming Liang. A langevinized ensemble kalman filter for large-
> scale static and dynamic learning. Statistica Sinica, 2023. doi: 10.5705/ss.202022.0172.
>
> ### reply to W2:
> #### Ans:
> The term "stage" refers to  a time step of the state space model, where a batch of training samples are collected based on the current value of $\theta$. In the revision, we will change it to ``step'' as used in some other RL papers.
>
> #### Ans:
> Yes, $x_t = (s_t, a_t, s_{t+1}, a_{t+1})$, which is the transition tuple excluding the reward $r_t$.
>
> #### Ans:
> In the revision, we will add explanations for all newly introduced notations. Specifically, $h(x_t,\theta_t)$ can be interpreted as a stochastic estimator of $E_{\rho} [r(s,a)|s,a]$ based on the new sample $s_{t+1}$ or  $(s_{t+1},a_{t+1})$, depending on our interest in $V$- or $Q$-functions.
>
> #### Ans:
> We use term to express the property of the proposed algorithm: It can potentially sample all optimal policies with correct proportions. We illustrates this property using Figure 4, which indicates the proposed algorithm can sample optimal policies in approximately correct proportions, while Adam fails to explore different optimal policies.
>
> ### reply to W3:
> The major goal of this research is to provide a method to quantify the uncertainty associated with the RL system. As demonstrated by Figure 2 and Figure 3, the proposed algorithms significantly outperform EKF and Adam in this perspective.
>
> As the second goal of this research, we use Figure A1 to show the proposed algorithm also performs very well in model training in terms of rewards. The proposed algorithm achieves comparable or better performance as Adam, a state-of-the-art algorithm in complex function optimization.
>
>
> ### reply to W4:
> #### Ans:
> It refers to a reward in the paper. As a function of the state variable $s$ and the action variable $a$, it is indeed a random variable in RL.
>
> #### Ans:
> The text will be changed accordingly.
>
> #### Ans:
> In the revision, we will particularly emphasize that (i) we provide a new formulation for RL, and (ii) provide an associated sampling algorithm and study its theoretical guarantee, enabling the uncertainty associated with the RL system to be properly quantified. However, the existing methods fail to do so.
> We note that our use of SGLD in this paper is new and should be classified as one of our proposed algorithms for RL. In the rebuttal, we have implemented an existing version of SGLD (Vanilla SGLD) for RL. on the mean square Bellman error, which cannot provide faithful confidence intervals as ours.
> The following figure shows the mean coverage rate of 95% prediction interval over 100 experiments.
> https://pasteboard.co/xHNnZ0kGEyoX.png
>
> #### Ans:
> The text will be changed accordingly.
>
> #### Ans:
> It will be expanded in the revision: The left plot shows the cumulative rewards obtained during the training process, the middle plot shows the testing performance without random exploration, and the right plot shows the performance of best model learned up to the current stage.

---

### Official Review · Reviewer_dX3U · 2023-11-03

**Soundness:** 2 fair
**Presentation:** 3 good
**Contribution:** 3 good
**Rating:** 6
**Confidence:** 3

**Summary:**

This paper delves into the stochastic nature in RL by introducing the Langevinized Kalman Temporal-Difference (LKTD) algorithm, which relies on an approximate sampling technique grounded in the Kalman filtering paradigm. By leveraging stochastic gradient Markov chain Monte Carlo (SGMCMC), this algorithm efficiently draws samples from the posterior distribution of deep neural network parameters, offering robust uncertainty quantification for deep RL.

**Strengths:**

Existing RL algorithms often treat the value function as a deterministic entity, overlooking the system's inherent stochastic nature. The Kalman Temporal Difference (KTD) framework in RL treats the value function as a random variable, aiming to track the policy learning process by formulating RL as a state space model. However, KTD-based techniques employing linearization methods become inefficient with high-dimensional models.

To improve the computational efficiency in deep RL, this paper reformulates RL with a new state space model, and adopts Langevinized ensemble Kalman filter (LEnKF) algorithm to ensure that the empirical distribution of the parameters in value function will converge to the desired target distribution. This reformulation enhances robustness by including prior information in the state evolution equation, robustifying performance, and handling large-scale neural network approximations. The proposed algorithm introduces computational efficiency, directly handling nonlinear functions without requiring linearization. It achieves memory efficiency by replacing the storage for the covariance matrix with particles, showcasing scalability for large-scale neural network approximations.

Providing the convergence guarantee in deep RL while incorporating approximate Bayesian inference can be challenging. This paper establishes promising performance guarantees in this regime.

**Weaknesses:**

1. The uncertainty quantification for Deep RL is not restricted to Kalman TD frameworks. In the literature of Bayesian RL, we do have works that adopt other approximate sampling techniques, e.g. ensembling in Bootstrapped DQN (Osband et.al. 2016), "Deep Exploration via Randomized Value Functions" (Osband et.al. 2019), and RLSVI with general function approximation (Ishfaq et.al., 2021). Although the above methods do not adopt MCMC methods as approximate sampling scheme, could authors compare the proposed method with the existing Bayesian RL methods, not just restricted to the Kalman TD context?

2. Though the authors provide a clear comparison to the existing Kalman TD frameworks,  it is desirable to see a more thorough related work study in Bayesian RL / uncertainty quantification in deep RL, which would be beneficial.

3. The proposed method requires a proper prior distribution to achieve the performance guarantee in LEnKF. It is unclear how the quality of the prior affects the performance through the current theorems. Is it possible to capture such quantity in the upper bounds provided in Theorem 1 or Theorem 2?

4. The experiment environments are relatively simple, especially for the control tasks. It is suggested authors consider more complicated locomotion tasks to demonstrate the effectiveness of the proposed method.

5. The proofs of Theorem 1 and 2 rely on a significant number of assumptions. It is suggested authors comment on whether those assumptions are imposed in order to achieve the convergence guarantee in sampling schemes, and how such assumptions can be satisfied in practice.

**Questions:**

1. Could authors comment on how many number of iterations are required in SGLD in order to achieve the convergence guarantee in Theorem 1? And how optimal is the convergence rate?

2. I am also curious if we adopt second-order MCMC methods such as SGHMC, could we obtain faster convergence with extra computational benefits? Is Theorem 1 generalizable to include the convergence guarantee of SGHMC methods? What can be potential challenges compared to SGLD?

---

> ### Author Response · Authors · 2023-11-23
> **Reply to Reviewer dX3U**
>
> ### reply to W1:
> In our most recent experiment, we compared our LKTD and the reformulated SGLD algorithms with DQN, BootDQN, and Vanilla SGMCMC algorithms, which do not incorporate our state space reformulation. The box plot of the coverage rate in Figure 1 (https://pasteboard.co/xHNnZ0kGEyoX.png) clearly shows that only the LKTD and reformulated SGLD algorithms achieve a reliable 95% prediction interval. In contrast, the other algorithms failed to converge to the true Q-value function. This failure is attributed to the semi-gradient derived from the DQN structure, which represents a biased gradient for the mean square Bellman equation.
>
> ### reply to W2:
> Bayesian DQN deploys Thompson sampling on the approximated posterior of the Q-function. However, the posterior distribution is conditional on the last hidden layer of the deep Q network. Since the network parameters are constantly changing throughout the training phase, it is unclear where the Q-function converges. On the other hand, LKTD guarantees the Q-network converges to a distribution around the optimal Q-function. Therefore, we can draw samples during the training phase and perform uncertainty quantification in a single experiment. Thanks for your advice, in the revision, we will include a more detailed explanation of the difference between our algorithm and related works.
> ### reply to W3:
> Thanks for your thoughtful question. The prior can generally affect
>     the constants $A$, $L_U$, $m_U$, $M_U$, and $B$, which are involved in the upper bound provided
>     in Theorem 1. However, as implied by equation (9) in Algorithm 1, the prior effect  on these constants is of the order $O(\frac{1}{\mathcal{N}})$, which can be diminished by increasing the pseudo population size $\mathcal{N}$.
> ### reply to W4:
> As suggested by another reviewer, we will include behavior suite environment in our revision.
>
> ### reply to W5:
> Thank you for your thoughtful question. In Lemma 1, we show that the Kalman temporal difference (KTD) framework used to solve the RL problem can be formulated as a preconditioned SGLD algorithm. This formulation is general without specific assumptions involved.
>     Then we establish the convergence of the preconditioned SGLD algorithm under the standard assumptions for stochastic gradient MCMC algorithms. We refer these assumptions to Raginsky et al. (2017) [1] (cited in the paper).  As explained below, these assumptions can be easily satisfied in practice.
> Assumption A1-(C1) concerns the stationarity of the Markov decision process (MDP), a regular condition for almost all RL studies.
>
> Assumption A1-(C3) basically requires the log-prior density function $\log \pi(\theta)$
>     and the neural network function $h(x,\theta)$ (with $\theta$ representing the weights
>     of the neural network) to be twice continuously differentiable. In this paper, we set $\pi(\theta)$
>     as  a mixture Gaussian distribution (see equation (A34)). Therefore, A1-(C3) is satisfied.
>
> Intuitively, the assumption A1-(C4) is to bound the dynamics inside a bounded domain; if $\theta_k$ is far away from the origin, then the dynamics are forced to get back around the origin. Although the neural network likelihood function is multimodal, the mixture Gaussian prior used in the paper forces the resulting posterior to be well behaved around the origin and thus to satisfy
>     A1-(C4) and A1-(C2).
>
> Assumption A1-(C5) regulates the variation of the stochastic gradient involved
>      at each iteration of the algorithm. This assumption can be satisfied by a pre-fixed value
>      of $\mathcal{N}$.
>
> Assumption A1-(C6) concerns the distribution of the initial weights of the neural network. It can be satisfied by initializing weights in a bounded space.
>
> Assumption A2 concerns the setting of the learning rate, which can be satisfied by setting $\epsilon_k=O(1/k^{\alpha})$ for some $1/2 < \alpha<1$. Assumption A3 is taken from Chen et al. (2015) [2] (cited in the paper), a regular assumption for test functions. Assumption A4 concerns the eigenvalues of the preconditioning matrix. It is automatically satisfied by the special structure
>      of $\Sigma_t= \frac{n}{\mathcal{N}}[I-\epsilon_t H_t^T(\epsilon_t H_t H_t^T+R_t)^{-1} H_t]$, following from an extension of Ostrowski's theorem [3].
>
> The above discussions will be incorporated into the revised manuscript.
>
> [1] Maxim Raginsky, Alexander Rakhlin, and Matus Telgarsky. Non-convex learning via
> stochastic gradient Langevin dynamics: a nonasymptotic analysis. In Proceedings of the
> 2017 Conference on Learning Theory, pp. 1674–1703, 2017.}
>
> [2] Changyou Chen, Nan Ding, and Lawrence Carin. On the convergence of stochastic gradient
> mcmc algorithms with high-order integrators. In Advances in Neural Information Processing
> Systems, pp. 2278–2286, 2015.
>
> [3] Nicholas J. Higham and Sheung Hun Cheng, Modifying the Inertia of Matrices Arising in Optimization, Linear Algebra Appl. 275–276, 261-279, 1998.

---

### Official Review · Reviewer_5LPR · 2023-11-04

**Soundness:** 2 fair
**Presentation:** 3 good
**Contribution:** 2 fair
**Rating:** 5
**Confidence:** 4

**Summary:**

This paper introduces a sampling algorithm called Langevinized Kalman Temporal-Difference (LKTD) for deep reinforcement learning. The algorithm leverages the Kalman filtering paradigm to efficiently draw samples from the approximate posterior distribution of deep neural network parameters, allowing for the quantification of uncertainties associated with the value function and model parameters. The LKTD algorithm improves the robustness and adaptability of reinforcement learning approaches.

**Strengths:**

The paper introduces the limitations of existing reinforcement learning algorithms that overlook the stochastic nature of the agent-environment interaction system. To address this, the authors propose a novel algorithm called Langevinized Kalman Temporal-Difference (LKTD) that leverages the Kalman filtering paradigm to draw samples from the posterior distribution of deep neural network parameters. The LKTD algorithm allows for quantifying uncertainties associated with the value function and model parameters, and enables monitoring of these uncertainties during policy updates throughout the training phase.

### Technical route of the article (step by step):

The LKTD algorithm is based on Langevin-Koopman Dynamic Mode Decomposition (LKTD) and Stochastic Gradient Langevin Dynamics (SGLD) sampler.
The mathematical equations and algorithms used in the LKTD algorithm are described, including the formulas for forecasting and analysis.
The convergence of the LKTD algorithm under the on-policy setting is discussed, along with the conditions and assumptions required for the convergence.
The choice of the pseudo-population size N and its impact on the system is explained.
The cooperation of the LKTD algorithm with a replay buffer is discussed, and a convergence theory for this scenario is presented.

**Weaknesses:**

1. Comparison with existing posterior sampling value-based algorithms for exploration is missing. Say ensemble sampling, Bootstrapped DQN or HyperDQN.
2. It is better to translate the theoretical guarantee to regret bound.
3. Comparison for uncertainty quantification in deep neural network is missing.
4. The empirical performance could be demonstrated through a wider range of benchmark problems, e.g. Arcade Learning benchmarks or behaviour suite.

**Questions:**

See weakness.

---

> ### Author Response · Authors · 2023-11-23
> **Reply to Reviewer 5LPR**
>
> ### reply to W1:
> We have implemented Bootstrapped DQN to construct 95\% prediction interval for the escape environment.  Due to the use of semi-gradient in DQN structure, the Q-functions construct by BootDQN is biased and thus the resulting confidence interval is less faithful.
> In figure 1 (https://pasteboard.co/xHNnZ0kGEyoX.png), the box plot of coverage rate shows that BootDQN failed to cover the true Q-value at the nominal 95\% level. In figure 2 (https://pasteboard.co/kBcqXjvKh6pP.png) , the MSE of Q-value estimates indicates that BootDQN exhibits a much larger bias compare to LKTD and SGLD (with pseudo population). In figure 3 (https://pasteboard.co/1mRkGnk2YJ0N.png), there is a clear decrease in interval length as the pseudo population size increases, which agrees with the statement in Remark 2. In our implementation for Bootstrap DQN, we set the learning rate to $10^{-4}$, and Bernoulli masking probability to 0.5. We demonstrated the results for two settings: head=5 and head=20.
>
> ### reply to W2:
> We don't use regret function as our objective function. Instead of minimizing the regret function, LKTD draws samples from the stationary distribution of Q-values. Hence, the performance of the algorithm is quantified by the bias between sample mean $\bar{\phi}_\mathcal{N}(\theta)$ and the population $\phi(\theta^*)$. As implied by equation (14) in Remark 1, the resulting regret is of the order $O(r_n^4 T/\mathcal{N}) + O(\sqrt{T})$, where $T$ denotes the number of time stages/steps.
>     Therefore, the proposed method obtains the ideal $O(\sqrt{T})$ error bounds by choosing a large value
>     of $\mathcal{N}$. In the revision, we will add a remark for this translation.
> ### reply to W3:
> In this rebuttal, we have included the results of SGLD and SGHMC, which were used to train deep neural networks but in the conventional  formulation. That is, we apply both algorithms directly to the mean square temporal difference loss function without state space reformulation.
>     \begin{equation}
>         \sum_{i=1}^n (Q(s,a|\theta) - r -\gamma \max_{a'} Q(s',a'|\theta))^2 + ||\theta||^2
>     \end{equation}
>     As shown by the plots given in the common reply, they could not produce correct coverage for Q-values. This experiment indicates the importance of our new formulation. For Vanilla SGLD and Vanilla SGHMC, both learning rate and temperature are set to $10^{-4}$ and 0.1 respectively. And the momentum coefficient for SGHMC is set to be 0.9.
> ### reply to W4:
>  We have applied LKTD to noisy environment in behavior suite: bandit_noise(https://pasteboard.co/LRriSJpY8lHy.png) and cart pole_noise(https://pasteboard.co/shFjTcVrL8jn.png), where average regret is used as a criterion of performance. However,  due to the nature of sampling algorithms, LKTD is expected to have slightly higher regret due to higher variation for uncertainty quantification. Hence, the coverage rate of the prediction interval is a better measurement for LKTD.

---

### Meta-Review · Area_Chair_jncq · 2023-12-07

**Metareview:**

a) The paper proposes a posterior-sampling based approach that allows for constructing intervals for value function prediction in deep RL, including a proof of convergence.

b) The reviewers unanimously rated the contribution of this paper as good; as one reviewer remarked, it is very difficult to prove convergence results in the deep RL setting.

c) The reviewers all identified lack of comparison to existing work in this area as an important weakness of the paper.

**Justification For Why Not Higher Score:**

The reviewers all gave borderline reviews

**Justification For Why Not Lower Score:**

Overall, this work seems to me to be a significant contribution to an important area.

---

### Decision · Program_Chairs · 2024-01-16

Accept (poster)